# Computational and Statistical Tradeoffs in Learning to Rank

**Ashish Khetan**    and    **Sewoong Oh**
Department of ISE, University of Illinois at Urbana-Champaign
Email: {khetan2,swoh}@illinois.edu

## Abstract

For massive and heterogeneous modern data sets, it is of fundamental interest to provide guarantees on the accuracy of estimation when computational resources are limited. In the application of learning to rank, we provide a hierarchy of rank-breaking mechanisms ordered by the complexity in thus generated sketch of the data. This allows the number of data points collected to be gracefully traded off against computational resources available, while guaranteeing the desired level of accuracy. Theoretical guarantees on the proposed generalized rank-breaking implicitly provide such trade-offs, which can be explicitly characterized under certain canonical scenarios on the structure of the data.

## 1   Introduction

In classical statistical inference, we are typically interested in characterizing how more data points improve the accuracy, with little restrictions or considerations on computational aspects of solving the inference problem. However, with massive growths of the amount of data available and also the complexity and heterogeneity of the collected data, computational resources, such as time and memory, are major bottlenecks in many modern applications. As a solution, recent advances in [7, 23, 8, 1, 16] introduce hierarchies of algorithmic solutions, ordered by the respective computational complexity, for several fundamental machine learning applications. Guided by sharp analyses on the sample complexity, these approaches provide theoretically sound guidelines that allow the analyst the flexibility to fall back to simpler algorithms to enjoy the full merit of the improved run-time.

Inspired by these advances, we study the time-data tradeoff in learning to rank. In many applications such as election, policy making, polling, and recommendation systems, we want to aggregate individual preferences to produce a global ranking that best represents the collective social preference. Learning to rank is a rank aggregation approach, which assumes that the data comes from a parametric family of choice models, and learns the parameters that determine the global ranking. Traditionally, each revealed preference is assumed to have one of the following three structures. *Pairwise comparison*, where one item is preferred over another, is common in sports and chess matches. *Best-out-of-$\kappa$ comparison*, where one is chosen among a set of $\kappa$ alternatives, is common in historical purchase data. *$\kappa$-way comparison*, where we observe a linear ordering of a set of $\kappa$ candidates, is used in some elections and surveys. For such traditional preferences, efficient schemes for learning to rank have been proposed, e.g. [12, 9]. However, modern data sets are unstructured and heterogeneous. This can lead to significant increase in the computational complexity, requiring exponential run-time in the size of the problem in the worst case [15].

To alleviate this computational challenge, we propose a hierarchy of estimators which we call *generalized rank-breaking*, ordered in increasing computational complexity and achieving increasing accuracy. The key idea is to break down the heterogeneous revealed preferences into simpler pieces of ordinal relations, and apply an estimator tailored for those simple structures treating each piece as independent. Several aspects of rank-breaking makes this problem interesting and challenging. A

priori, it is not clear which choices of the simple ordinal relations are rich enough to be statistically efficient and yet lead to tractable estimators. Even if we identify which ordinal relations to extract, the ignored correlations among those pieces can lead to an inconsistent estimate, unless we choose carefully which pieces to include and which to omit in the estimation. We further want sharp analysis on the sample complexity, which reveals how computational and statistical efficiencies trade off. We would like to address all these challenges in providing generalized rank-breaking methods.

**Problem formulation.** We study the problem of aggregating ordinal data based on users' preferences that are expressed in the form of *partially ordered sets (poset)*. A poset is a collection of ordinal relations among items. For example, consider a poset $\{(i_6 \prec \{i_5, i_4\}), (i_5 \prec i_3), (\{i_3, i_4\} \prec \{i_1, i_2\})\}$ over items $\{i_1, \ldots, i_6\}$, where $(i_6 \prec \{i_5, i_4\})$ indicates that item $i_5$ and $i_4$ are both preferred over item $i_6$. Such a relation is extracted from, for example, the user giving a 2-star rating to $i_5$ and $i_4$ and a 1-star to $i_6$. Assuming that the revealed preference is consistent, a poset can be represented as a directed acyclic graph (DAG) $\mathcal{G}_j$ as below.

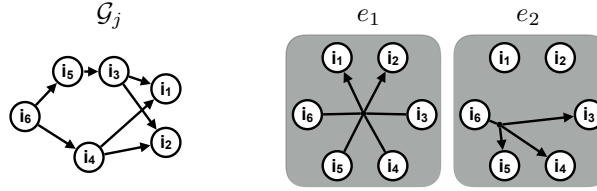

Figure 1: An example of $\mathcal{G}_j$ for user $j$'s consistent poset, and two rank-breaking hyper edges extracted from it: $e_1 = (\{i_6, i_5, i_4, i_3\} \prec \{i_2, i_1\})$ and $e_2 = (\{i_6\} \prec \{i_5, i_4, i_3\})$.

We assume that each user $j$ is presented with a subset of items $S_j$, and independently provides her ordinal preference in the form of a poset, where the ordering is drawn from the Plackett-Luce (PL) model. The PL model is a popular choice model from operations research and psychology, used to model how people make choices under uncertainty. It is a special case of *random utility models*, where each item $i$ is parametrized by a latent true utility $\theta_i \in \mathbb{R}$. When offered with $S_j$, the user samples the perceived utility $U_i$ for each item independently according to $U_i = \theta_i + Z_i$, where $Z_i$'s are i.i.d. noise. In particular, the PL model assumes $Z_i$'s follow the standard Gumbel distribution. Although statistical and computational tradeoff has been studied under Mallows models [6] or stochastically transitive models [22], the techniques we develop are different and have a potential to generalize to analyze more general class of random utility models. The observed poset is a partial observation of the ordering according to this perceived utilities.

The particular choice of the Gumbel distribution has several merits, largely stemming from the fact that the Gumbel distribution has a log-concave pdf and is inherently memoryless. In our analyses, we use the log-concavity to show that our proposed algorithm is a concave maximization (Remark 2.1) and the memoryless property forms the basis of our rank-breaking idea. Precisely, the PL model is statistically equivalent to the following procedure. Consider a ranking as a mapping from a rank to an item, i.e. $\sigma_j : [|S_j|] \to S_j$. It can be shown that the PL model is generated by first independently assigning each item $i \in S_j$ an unobserved value $Y_i$, exponentially distributed with mean $e^{-\theta_i}$, and the resulting ranking $\sigma_j$ is inversely ordered in $Y_i$'s so that $Y_{\sigma_j(1)} \leq Y_{\sigma_j(2)} \leq \cdots \leq Y_{\sigma_j(|S_j|)}$.

This inherits the memoryless property of exponential variables, such that $\mathbb{P}(Y_1 < Y_2 < Y_3) = \mathbb{P}(Y_1 < \{Y_2, Y_3\}) \mathbb{P}(Y_2 < Y_3)$, leading to a simple interpretation of the PL model as sequential choices: $\mathbb{P}(i_3 \prec i_2 \prec i_1) = \mathbb{P}(\{i_3, i_2\} \prec i_1) \mathbb{P}(i_3 \prec i_2) = (e^{\theta_{i_1}}/(e^{\theta_{i_1}} + e^{\theta_{i_2}} + e^{\theta_{i_3}})) \times (e^{\theta_{i_2}}/(e^{\theta_{i_2}} + e^{\theta_{i_3}}))$. In general, we have $\mathbb{P}[\sigma_j] = \prod_{i=1}^{|S_j|-1} (e^{\theta^*_{\sigma_j(i)}})/(\sum_{i'=i}^{|S_j|} e^{\theta^*_{\sigma_j(i')}})$. We assume that the true utility $\theta^* \in \Omega_b$ where $\Omega_b = \{\theta \in \mathbb{R}^d | \sum_{i \in [d]} \theta_i = 0, |\theta_i| \leq b \text{ for all } i \in [d]\}$. Notice that centering of $\theta$ ensures its uniqueness as PL model is invariant under shifting of $\theta$. The bound $b$ on $\theta_i$ is written explicitly to capture the dependence in our main results.

We denote a set of $n$ users by $[n] = \{1, \ldots, n\}$ and the set of $d$ items by $[d]$. Let $\mathcal{G}_j$ denote the DAG representation of the poset provided by the user $j$ over $S_j \subseteq [d]$ according to the PL model with weights $\theta^*$. The maximum likelihood estimate (MLE) maximizes the sum of all possible rankings

that are consistent with the observed $\mathcal{G}_j$ for each $j$:

$$\widehat{\theta} \quad \in \quad \arg\max_{\theta \in \Omega_b} \left\{ \sum_{j=1}^{n} \log \left( \sum_{\sigma \in \mathcal{G}_j} \mathbb{P}_\theta[\sigma] \right) \right\}, \tag{1}$$

where we slightly abuse the notation $\mathcal{G}_j$ to denote the set of all rankings $\sigma$ that are consistent with the observation. When $\mathcal{G}_j$ has a *traditional* structure as explained earlier in this section, then the optimization is a simple multinomial logit regression, that can be solved efficiently with off-the-shelf convex optimization tools [12]. For general posets, it can be shown that the above optimization is a concave maximization, using similar techniques as Remark 2.1. However, the summation over rankings in $\mathcal{G}_j$ can involve number of terms super exponential in the size $|S_j|$, in the worst case. This renders MLE intractable and impractical.

**Pairwise rank-breaking.** A common remedy to this computational blow-up is to use rank-breaking. Rank-breaking traditionally refers to *pairwise rank-breaking*, where a bag of all the pairwise comparisons is extracted from observations $\{\mathcal{G}_j\}_{j \in [n]}$ and is applied to estimators that are tailored for pairwise comparisons, treating each paired outcome as independent. This is one of the motivations behind the algorithmic advances in learning from pairwise comparisons [19, 21, 17].

It is computationally efficient to apply maximum likelihood estimator assuming independent pairwise comparisons, which takes $O(d^2)$ operations to evaluate. However, this computational gain comes at the cost of statistical efficiency. It is known from [4] that if we include all paired comparisons, then the resulting estimate can be statistically inconsistent due to the ignored correlations among the paired orderings, even with infinite samples. In the example from Figure 1, there are 12 paired relations: $(i_6 \prec i_5), (i_6 \prec i_4), (i_6 \prec i_3), \ldots, (i_3 \prec i_1), (i_4 \prec i_1)$. In order to get a consistent estimate, [4] provides a rule for choosing which pairs to include, and [15] provides an estimator that optimizes how to weigh each of those chosen pairs to get the best finite sample complexity bound. However, such a consistent pairwise rank-breaking results in throwing away many of the ordered relations, resulting in significant loss in accuracy. For example, none of the pairwise orderings can be used from $\mathcal{G}_j$ in the example, without making the estimator inconsistent [3]. Whether we include all paired comparisons or only a subset of consistent ones, there is a significant loss in accuracy as illustrated in Figure 2. For the precise condition for consistent rank-breaking we refer to [3, 4, 15].

The state-of-the-art approaches operate on either one of the two extreme points on the computational and statistical trade-off. The MLE in (1) requires $O(\sum_{j \in [n]} |S_j|!)$ summations to just evaluate the objective function, in the worst case. On the other hand, the pairwise rank-breaking requires only $O(d^2)$ summations, but suffers from significant loss in the sample complexity. Ideally, we would like to give the analyst the flexibility to choose a target computational complexity she is willing to tolerate, and provide an algorithm that achieves the optimal trade-off at any operating point.

**Contribution.** We introduce a novel *generalized rank-breaking* that bridges the gap between MLE and pairwise rank-breaking. Our approach allows the user the freedom to choose the level of computational resources to be used, and provides an estimator tailored for the desired complexity. We prove that the proposed estimator is tractable and consistent, and provide an upper bound on the error rate in the finite sample regime. The analysis explicitly characterizes the dependence on the topology of the data. This in turn provides a guideline for designing surveys and experiments in practice, in order to maximize the sample efficiency. We provide numerical experiments confirming the theoretical guarantees.

## 2 Generalized rank-breaking

Given $\mathcal{G}_j$'s representing the users' preferences, *generalized rank-breaking* extracts a set of ordered relations and applies an estimator treating each ordered relation as independent. Concretely, for each $\mathcal{G}_j$, we first extract a maximal ordered partition $\mathcal{P}_j$ of $S_j$ that is consistent with $\mathcal{G}_j$. An ordered partition is a partition with a linear ordering among the subsets, e.g. $\mathcal{P}_j = (\{i_6\} \prec \{i_5, i_4, i_3\} \prec \{i_2, i_1\})$ for $\mathcal{G}_j$ from Figure 1. This is maximal, since we cannot further partition any of the subsets without creating artificial ordered relations that are not present in the original $\mathcal{G}_j$.

The extracted ordered partition is represented by a directed hypergraph $G_j(S_j, E_j)$, which we call a *rank-breaking graph*. Each edge $e = (B(e), T(e)) \in E_j$ is a directed hyper edge from a subset of nodes $B(e) \subseteq S_j$ to another subset $T(e) \subseteq S_j$. The number of edges in $E_j$ is $|\mathcal{P}_j| - 1$

where $|\mathcal{P}_j|$ is the number of subsets in the partition. For each subset in $\mathcal{P}_j$ except for the least preferred subset, there is a corresponding edge whose *top-set* $T(e)$ is the subset, and the *bottom-set* $B(e)$ is the set of all items less preferred than $T(e)$. In Figure 1, for $E_j = \{e_1, e_2\}$ we show $e_1 = (B(e_1), T(e_1)) = (\{i_6, i_5, i_4, i_3\}, \{i_2, i_1\})$ and $e_2 = (B(e_2), T(e_2)) = (\{i_6\}, \{i_5, i_4, i_3\})$ extracted from $\mathcal{G}_j$. Denote the probability that $T(e)$ is preferred over $B(e)$ when $T(e) \cup B(e)$ is offered as

$$\mathbb{P}_\theta(e) = \mathbb{P}_\theta\big(B(e) \prec T(e)\big) = \sum_{\sigma \in \Lambda_{T(e)}} \frac{\exp\left(\sum_{c=1}^{|T(e)|} \theta_{\sigma(c)}\right)}{\prod_{u=1}^{|T(e)|} \left(\sum_{c'=u}^{|T(e)|} \exp\left(\theta_{\sigma(c')}\right) + \sum_{i \in B(e)} \exp\left(\theta_i\right)\right)} \quad (2)$$

which follows from the definition of the PL model, where $\Lambda_{T(e)}$ is the set of all rankings over $T(e)$. The computational complexity of evaluating this probability is dominated by the size of the *top-set* $|T(e)|$, as it involves $(|T(e)|!)$ summations. We let the analyst choose the order $M \in \mathbb{Z}^+$ depending on how much computational resource is available, and only include those edges with $|T(e)| \leq M$ in the following step. We apply the MLE for comparisons over paired subsets, assuming all rank-breaking graphs are independently drawn. Precisely, we propose *order-$M$ rank-breaking estimate*, which is the solution that maximizes the log-likelihood under the independent assumption:

$$\widehat{\theta} \in \arg\max_{\theta \in \Omega_b} \mathcal{L}_{\mathrm{RB}}(\theta) \ , \text{ where } \quad \mathcal{L}_{\mathrm{RB}}(\theta) = \sum_{j \in [n]} \sum_{e \in E_j : |T(e)| \leq M} \log \mathbb{P}_\theta(e) \ . \quad (3)$$

In a special case when $M = 1$, this can be transformed into the traditional pairwise rank-breaking, where $(i)$ this is a concave maximization; $(ii)$ the estimate is (asymptotically) unbiased and consistent [3, 4]; and $(iii)$ and the finite sample complexity have been analyzed [15]. Although, this order-1 rank-breaking provides a significant gain in computational efficiency, the information contained in higher-order edges are unused, resulting in a significant loss in sample efficiency.

We provide the analyst the freedom to choose the computational complexity he/she is willing to tolerate. However, for general $M$, it has not been known if the optimization in (3) is tractable and/or if the solution is consistent. Since $\mathbb{P}_\theta(B(e) \prec T(e))$ as explicitly written in (2) is a sum of log-concave functions, it is not clear if the sum is also log-concave. Due to the ignored dependency in the formulation (3), it is not clear if the resulting estimate is consistent. We first establish that it is a concave maximization in Remark 2.1, then prove consistency in Remark 2.2, and provide a sharp analysis of the performance in the finite sample regime, characterizing the trade-off between computation and sample size in Section 4. We use the Random Utility Model (RUM) interpretation of the PL model to prove concavity. We refer to Appendix A in the supplementary material for a proof.

**Remark 2.1.** $\mathcal{L}_{\mathrm{RB}}(\theta)$ *is concave in* $\theta \in \mathbb{R}^d$.

For consistency, we consider a simple but canonical scenario for sampling ordered relations. However, we study a general sampling scenario, when we analyze the order-$M$ estimator in the finite sample regime in Section 4. Following is the canonical sampling scenario. There is a set of $\tilde{\ell}$ integers $(\tilde{m}_1, \ldots, \tilde{m}_{\tilde{\ell}})$ whose sum is strictly less than $d$. A new arriving user is presented with all $d$ items and is asked to provide her top $\tilde{m}_1$ items as an unordered set, and then the next $\tilde{m}_2$ items, and so on. This is sampling from the PL model and observing an ordered partition with $(\tilde{\ell} + 1)$ subsets of sizes $\tilde{m}_a$'s, and the last subset includes all remaining items. We apply the generalized rank-breaking to get rank-breaking graphs $\{G_j\}$ with $\tilde{\ell}$ edges each, and order-$M$ estimate is computed. We show that this is consistent, i.e. asymptotically unbiased in the limit of the number of users $n$. A proof is provided in the supplementary material.

**Remark 2.2.** *Under the* PL *model and the above sampling scenario, the order-$M$ rank-breaking estimate* $\widehat{\theta}$ *in* (3) *is consistent for all choices of* $M \geq \min_{a \in \tilde{\ell}} \tilde{m}_a$.

Figure 2 (left) illustrates the trade-off between run-time and sample size necessary to achieve a fixed accuracy: MSE$\leq 0.3d^2 \times 10^{-6}$. In the middle panel, we show the accuracy-sample tradeoff for increasing computation $M$ on the same data. We fix $d = 256$, $\tilde{\ell} = 5$, $\tilde{m}_a = a$ for $a \in \{1, 2, 3, 4, 5\}$, and sample posets from the canonical scenario, except that each user is presented $\kappa = 32$ random items. The PL weights are chosen i.i.d. $U[-2, 2]$. On the right panel, we let $\tilde{m}_a = 3$ for all $a \in [\tilde{\ell}]$ and vary $\tilde{\ell}$. We compare GRB with $M = 3$ to PRB, and an oracle estimator who knows the exact ordering among those top three items and runs MLE.

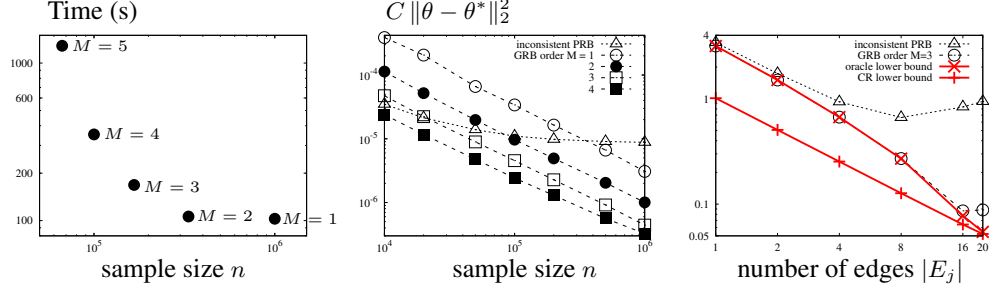

Figure 2: The time-data trade-off for fixed accuracy (left) and accuracy improvement for increased computation $M$ (middle). Generalized Rank-Breaking (GRB) achieves the oracle lower bound and significantly improves upon Pairwise Rank-Breaking (PRB) (right).

**Notations.** Given rank-breaking graphs $\{G_j(S_j, E_j)\}_{j \in [n]}$ extracted from the posets $\{\mathcal{G}_j\}$, we first define the order $M$ rank-breaking graphs $\{G_j^{(M)}(S_j, E_j^{(M)})\}$, where $E_j^{(M)}$ is a subset of $E_j$ that includes only those edges $e_j \in E_j$ with $|T(e_j)| \leq M$. This represents those edges that are included in the estimation for a choice of $M$. For finite sample analysis, the following quantities capture how the error depends on the topology of the data collected. Let $\kappa_j \equiv |S_j|$ and $\ell_j \equiv |E_j^{(M)}|$. We index each edge $e_j$ in $E_j^{(M)}$ by $a \in [\ell_j]$ and define $m_{j,a} \equiv |T(e_{j,a})|$ for the $a$-th edge of the $j$-th rank-breaking graph and $r_{j,a} \equiv |T(e_{j,a})| + |B(e_{j,a})|$. Note that, we use tilde in subscript with $m_{j,a}$ and $\ell_j$ when $M$ is equal to $S_j$. That is $\tilde{\ell}_j$ is the number of edges in $E_j$ and $\tilde{m}_{j,a}$ is the size of the top-sets in those edges. We let $p_j \equiv \sum_{a \in [\ell_j]} m_{j,a}$ denote the effective sample size for the observation $G_j^{(M)}$, such that the total effective sample size is $\sum_{j \in [n]} p_j$. Notice that although we do not explicitly write the dependence on $M$, all of the above quantities implicitly depend on the choice of $M$.

## 3 Comparison graph

The analysis of the optimization in (3) shows that, with high probability, $\mathcal{L}_{\mathrm{RB}}(\theta)$ is strictly concave with $\lambda_2(H(\theta)) \leq -C_b \gamma_1 \gamma_2 \gamma_3 \lambda_2(L) < 0$ for all $\theta \in \Omega_b$ (Lemma C.3), and the gradient is also bounded with $\|\nabla \mathcal{L}_{\mathrm{RB}}(\theta^*)\| \leq C_b' \gamma_2^{-1/2} (\sum_j p_j \log d)^{1/2}$ (Lemma C.2). the quantities $\gamma_1$, $\gamma_2$, $\gamma_3$, and $\lambda_2(L)$, to be defined shortly, represent the topology of the data. This leads to Theorem 4.1:

$$\|\widehat{\theta} - \theta^*\|_2 \ \leq \ \frac{2\|\nabla \mathcal{L}_{\mathrm{RB}}(\theta^*)\|}{-\lambda_2(H(\theta))} \ \leq \ C_b'' \frac{\sqrt{\sum_j p_j \log d}}{\gamma_1 \gamma_2^{3/2} \gamma_3 \lambda_2(L)} \ , \tag{4}$$

where $C_b$, $C_b'$, and $C_b''$ are constants that only depend on $b$, and $\lambda_2(H(\theta))$ is the second largest eigenvalue of a negative semidefinite Hessian matrix $H(\theta)$ of $\mathcal{L}_{\mathrm{RB}}(\theta)$. Recall that $\theta^\top \mathbf{1} = 0$ since we restrict our search in $\Omega_b$. Hence, the error depends on $\lambda_2(H(\theta))$ instead of $\lambda_1(H(\theta))$ whose corresponding eigen vector is the all-ones vector. We define a *comparison graph* $\mathcal{H}([d], E)$ as a weighted undirected graph with weights $A_{ii'} = \sum_{j \in [n]: i, i' \in S_j} p_j / (\kappa_j(\kappa_j - 1))$. The corresponding graph Laplacian is defined as:

$$L \ \equiv \ \sum_{j=1}^{n} \frac{p_j}{\kappa_j(\kappa_j - 1)} \sum_{i < i' \in S_j} (e_i - e_{i'})(e_i - e_{i'})^\top . \tag{5}$$

It is immediate that $\lambda_1(L) = 0$ with $\mathbf{1}$ as the eigenvector. There are remaining $d - 1$ eigenvalues that sum to $\mathrm{Tr}(L) = \sum_j p_j$. The rescaled $\lambda_2(L)$ and $\lambda_d(L)$ capture the dependency on the topology:

$$\alpha \equiv \frac{\lambda_2(L)(d - 1)}{\mathrm{Tr}(L)} \ , \qquad \beta \equiv \frac{\mathrm{Tr}(L)}{\lambda_d(L)(d - 1)} . \tag{6}$$

In an ideal case where the graph is well connected, then the spectral gap of the Laplacian is large. This ensures all eigenvalues are of the same order and $\alpha = \beta = \Theta(1)$, resulting in a smaller error

rate. The concavity of $\mathcal{L}_{\mathrm{RB}}(\theta)$ also depends on the following quantities. We discuss the role of the topology in Section 4. Note that the quantities defined in this section implicitly depend on the choice of $M$, which controls the necessary computational power, via the definition of the rank-breaking $\{G_{j,a}\}$. We define the following quantities that control our upper bound. $\gamma_1$ incorporates asymmetry in probabilities of items being ranked at different positions depending upon their weight $\theta_i^*$. It is $1$ for $b = 0$ that is when all the items have same weight, and decreases exponentially with increase in $b$. $\gamma_2$ controls the range of the size of the top-set with respect to the size of the bottom-set for which the error decays with the rate of $1/($size of the top-set$)$. The dependence in $\gamma_3$ and $\nu$ are due to weakness in the analysis, and ensures that the Hessian matrix is strictly negative definite.

$$\gamma_1 \equiv \min_{j,a}\left\{\left(\frac{r_{j,a}-m_{j,a}}{\kappa_j}\right)^{2e^{2b}-2}\right\}, \quad \gamma_2 \equiv \min_{j,a}\left\{\left(\frac{r_{j,a}-m_{j,a}}{r_{j,a}}\right)^2\right\}, \text{ and} \tag{7}$$

$$\gamma_3 \equiv 1 - \max_{j,a}\left\{\frac{4e^{16b}}{\gamma_1}\frac{m_{j,a}^2 r_{j,a}^2 \kappa_j^2}{(r_{j,a}-m_{j,a})^5}\right\}, \quad \nu \equiv \max_{j,a}\left\{\frac{m_{j,a}\kappa_j^2}{(r_{j,a}-m_{j,a})^2}\right\}. \tag{8}$$

# 4 Main Results

We present main theoretical analyses and numerical simulations confirming the theoretical predictions.

## 4.1 Upper bound on the achievable error

We provide an upper bound on the error for the order-$M$ rank-breaking approach, showing the explicit dependence on the topology of the data. We assume each user provides a partial ranking according to his/her ordered partitions. Precisely, we assume that the set of offerings $S_j$, the number of subsets $(\tilde{\ell}_j + 1)$, and their respective sizes $(\tilde{m}_{j,1}, \ldots, \tilde{m}_{j,\tilde{\ell}_j})$ are *predetermined*. Each user randomly draws a ranking of items from the PL model, and provides the partial ranking of the form $(\{i_6\} \prec \{i_5, i_4, i_3\} \prec \{i_2, i_1\})$ in the example in Figure 1. For a choice of $M$, the order-$M$ rank-breaking graph is extracted from this data. The following theorem provides an upper bound on the achieved error, and a proof is provided in the supplementary material.

**Theorem 4.1.** *Suppose there are $n$ users, $d$ items parametrized by $\theta^* \in \Omega_b$, and each user $j \in [n]$ is presented with a set of offerings $S_j \subseteq [d]$ and provides a partial ordering under the PL model. For a choice of $M \in \mathbb{Z}^+$, if $\gamma_3 > 0$ and the effective sample size $\sum_{j=1}^n p_j$ is large enough such that*

$$\sum_{j=1}^n p_j \geq \frac{2^{14}e^{20b}\nu^2}{(\alpha\gamma_1\gamma_2\gamma_3)^2\beta}\frac{p_{\max}}{\kappa_{\min}}d\log d, \tag{9}$$

*where $b \equiv \max_i|\theta_i^*|$ is the dynamic range, $p_{\max} = \max_{j\in[n]} p_j$, $\kappa_{\min} = \min_{j\in[n]} \kappa_j$, $\alpha$ is the (rescaled) spectral gap, $\beta$ is the (rescaled) spectral radius in (6), and $\gamma_1$, $\gamma_2$, $\gamma_3$, and $\nu$ are defined in (7) and (8), then the generalized rank-breaking estimator in (3) achieves*

$$\frac{1}{\sqrt{d}}\|\widehat{\theta} - \theta^*\| \leq \frac{40e^{7b}}{\alpha\gamma_1\gamma_2^{3/2}\gamma_3}\sqrt{\frac{d\log d}{\sum_{j=1}^n\sum_{a=1}^{\ell_j} m_{j,a}}}, \tag{10}$$

*with probability at least $1 - 3e^3 d^{-3}$. Moreover, for $M \leq 3$ the above bound holds with $\gamma_3$ replaced by one, giving a tighter result.*

Note that the dependence on the choice of $M$ is not explicit in the bound, but rather is implicit in the construction of the comparison graph and the number of effective samples $N = \sum_j\sum_{a\in[\ell_j]} m_{j,a}$.

In an ideal case, $b = O(1)$ and $m_{j,a} = O(r_{j,a}^{1/2})$ for all $(j,a)$ such that $\gamma_1, \gamma_2$ are finite. further, if the spectral gap is large such that $\alpha > 0$ and $\beta > 0$, then Equation (10) implies that we need the effective sample size to scale as $O(d\log d)$, which is only a logarithmic factor larger than the number of parameters. In this ideal case, there exist universal constants $C_1, C_2$ such that if $m_{j,a} < C_1\sqrt{r_{j,a}}$ and $r_{j,a} > C_2\kappa_j$ for all $\{j,a\}$, then the condition $\gamma_3 > 0$ is met. Further, when $r_{j,a} = O(\kappa_{j,a})$, $\max \kappa_{j,a}/\kappa_{j',a'} = O(1)$, and $\max p_{j,a}/p_{j',a'} = O(1)$, then condition on the effective sample size is met with $\sum_j p_j = O(d\log d)$. We believe that dependence in $\gamma_3$ is weakness of our analysis and there is no dependence as long as $m_{j,a} < r_{j,a}$.

## 4.2 Lower bound on computationally unbounded estimators

Recall that $\tilde{\ell}_j \equiv |E_j|$, $\tilde{m}_{j,a} = |T(e_a)|$ and $\tilde{r}_{j,a} = |T(e_a) \cup B(e_a)|$ when $M = S_j$. We prove a fundamental lower bound on the achievable error rate that holds for any *unbiased* estimator even with no restrictions on the computational complexity. For each $(j, a)$, define $\eta_{j,a}$ as

$$\eta_{j,a} = \sum_{u=0}^{\tilde{m}_{j,a}-1} \left( \frac{1}{\tilde{r}_{j,a} - u} + \frac{u(\tilde{m}_{j,a} - u)}{\tilde{m}_{j,a}(\tilde{r}_{j,a} - u)^2} \right) + \sum_{u < u' \in [\tilde{m}_{j,a}-1]} \frac{2u}{\tilde{m}_{j,a}(\tilde{r}_{j,a} - u)} \frac{\tilde{m}_{j,a} - u'}{\tilde{r}_{j,a} - u'} \quad (11)$$

$$= \tilde{m}_{j,a}^2 / (3\tilde{r}_{j,a}) + O(\tilde{m}_{j,a}^3 / \tilde{r}_{j,a}^2). \quad (12)$$

**Theorem 4.2.** *Let $\mathcal{U}$ denote the set of all unbiased estimators of $\theta^*$ that are centered such that $\widehat{\theta}\mathbf{1} = 0$, and let $\mu = \max_{j \in [n], a \in [\tilde{\ell}_j]}\{\tilde{m}_{j,a} - \eta_{j,a}\}$. For all $b > 0$,*

$$\inf_{\widehat{\theta} \in \mathcal{U}} \sup_{\theta^* \in \Omega_b} \mathbb{E}[\|\widehat{\theta} - \theta^*\|^2] \geq \max \left\{ \frac{(d-1)^2}{\sum_{j=1}^n \sum_{a=1}^{\tilde{\ell}_j}(\tilde{m}_{j,a} - \eta_{j,a})} , \frac{1}{\mu} \sum_{i=2}^d \frac{1}{\lambda_i(L)} \right\} . \quad (13)$$

The proof relies on the Cramer-Rao bound and is provided in the supplementary material. Since $\eta_{j,a}$'s are non-negative, the mean squared error is lower bounded by $(d-1)^2/N$, where $N = \sum_j \sum_{a \in \tilde{\ell}_j} \tilde{m}_{j,a}$ is the effective sample size. Comparing it to the upper bound in (10), this is tight up to a logarithmic factor when $(a)$ the topology of the data is well-behaved such that all respective quantities are finite; and $(b)$ there is no limit on the computational power and $M$ can be made as large as we need. The bound in Eq. (13) further gives a tighter lower bound, capturing the dependency in $\eta_{j,a}$'s and $\lambda_i(L)$'s. Considering the first term, $\eta_{j,a}$ is larger when $\tilde{m}_{j,a}$ is close to $\tilde{r}_{j,a}$, giving a tighter bound. The second term in (13) implies we get a tighter bound when $\lambda_2(L)$ is smaller.

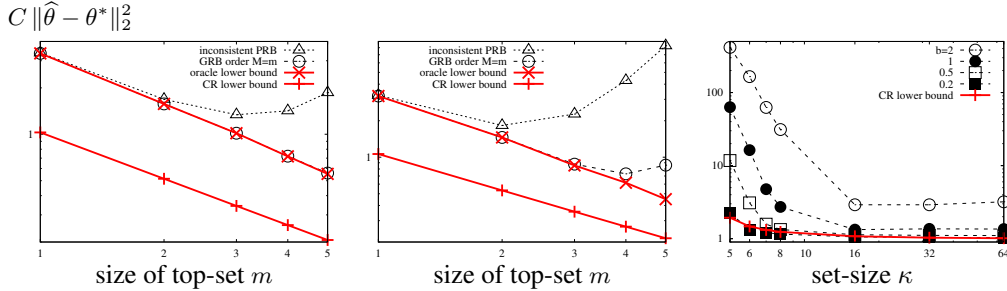

Figure 3: Accuracy degrades as $(\kappa - m)$ gets small and as the dynamic range $b$ gets large.

In Figure 3 left and middle panel, we compare performance of our algorithm with pairwise breaking, Cramer Rao lower bound and oracle MLE lower bound. We fix $d = 512$, $n = 10^5$, $\theta^*$ chosen i.i.d. uniformly over $[-2, 2]$. Oracle MLE knows relative ordering of items in all the top-sets $T(e)$ and hence is strictly better than the GRB. We fix $\tilde{\ell} = \ell = 1$ that is $r = \kappa$, and vary $m$. In the left panel, we fix $\kappa = 32$ and in the middle panel, we fix $\kappa = 16$. Perhaps surprisingly, GRB matches with the oracle MLE which means relative ordering of top-$m$ items among themselves is statistically insignificant when $m$ is sufficiently small in comparison to $\kappa$. For $\kappa = 16$, as $m$ gets large, the error starts to increase as predicted by our analysis. The reason is that the quantities $\gamma_1$ and $\gamma_2$ gets smaller as $m$ increases, and the upper bound increases consequently. In the right panel, we fix $m = 4$. When $\kappa$ is small, $\gamma_2$ is small, and hence error is large; when $b$ is large $\gamma_1$ is exponentially small, and hence error is significantly large. This is different from learning Mallows models, where peaked distributions are easier to learn [2], and is related to the fact that we are not only interested in recovering the (ordinal) ranking but also the (cardinal) weight.

## 4.3 Computational and statistical tradeoff

For estimators with limited computational power, however, the above lower bound fails to capture the dependency on the allowed computational power. Understanding such fundamental trade-offs is a challenging problem, which has been studied only in a few special cases, e.g. planted clique problem

[10, 18]. This is outside the scope of this paper, and we instead investigate the trade-off achieved by the proposed rank-breaking approach. When we are limited on computational power, Theorem 4.1 implicitly captures this dependence when order-$M$ rank-breaking is used. The dependence is captured indirectly via the resulting rank-breaking $\{G_{j,a}\}_{j\in[n],a\in[\ell_j]}$ and the topology of it. We make this trade-off explicit by considering a simple but canonical example. Suppose $\theta^* \in \Omega_b$ with $b = O(1)$. Each user gives an i.i.d. partial ranking, where all items are offered and the partial ranking is based on an ordered partition with $\tilde{\ell}_j = \lfloor \sqrt{2c}d^{1/4} \rfloor$ subsets. The top subset has size $\tilde{m}_{j,1} = 1$, and the $a$-th subset has size $\tilde{m}_{j,a} = a$, up to $a < \tilde{\ell}_j$, in order to ensure that they sum at most to $c\sqrt{d}$ for sufficiently small positive constant $c$ and the condition on $\gamma_3 > 0$ is satisfied. The last subset includes all the remaining items in the bottom, ensuring $\tilde{m}_{j,\tilde{\ell}_j} \geq d/2$ and $\gamma_1, \gamma_2$ and $\nu$ are all finite.

**Computation.** For a choice of $M$ such that $M \leq \ell_j - 1$, we consider the computational complexity in evaluating the gradient of $\mathcal{L}_{\mathrm{RB}}$, which scales as $T_M = \sum_{j\in[n]} \sum_{a\in[M]} (m_{j,a}!) r_{j,a} = O(M! \times dn)$. Note that we find the MLE by solving a convex optimization problem using first order methods, and detailed analysis of the convergence rate and the complexity of solving general convex optimizations is outside the scope of this paper.

**Sample.** Under the canonical setting, for $M \leq \ell_j - 1$, we have $L = M(M+1)/(2d(d-1))(\mathbb{I}-\mathbf{1}\mathbf{1}^\top)$. This complete graph has the largest possible spectral gap, and hence $\alpha > 0$ and $\beta > 0$. Since the effective samples size is $\sum_{j,a} \tilde{m}_{j,a}\mathbb{I}\{\tilde{m}_{j,a} \leq M\} = nM(M+1)/2$, it follows from Theorem 4.1 that the (rescaled) root mean squared error is $O(\sqrt{(d\log d)/(nM^2)})$. In order to achieve a target error rate of $\varepsilon$, we need to choose $M = \Omega((1/\varepsilon)\sqrt{(d\log d)/n})$. The resulting trade-off between run-time and sample to achieve root mean squared error $\varepsilon$ is $T(n) \propto (\lceil(1/\varepsilon)\sqrt{(d\log d)/n}\rceil)!dn$. We show numerical experiment under this canonical setting in Figure 2 (left) with $d = 256$ and $M \in \{1,2,3,4,5\}$, illustrating the trade-off in practice.

## 4.4  Real-world data sets

On sushi preferences [14] and jester dataset [11], we improve over pairwise breaking and achieves same performance as the oracle MLE. Full rankings over $\kappa = 10$ types of sushi are randomly chosen from $d = 100$ types of sushi are provided by $n = 5000$ individuals. As the ground truth $\theta^*$, we use the ML estimate of PL weights over the entire data. In Figure 4, left panel, for each $m \in \{3, 4, 5, 6, 7\}$, we remove the known ordering among the top-$m$ and bottom-$(10 - m)$ sushi in each set, and run our estimator with one breaking edge between top-$m$ and bottom-$(10 - m)$ items. We compare our algorithm with inconsistent pairwise breaking (using optimal choice of parameters from [15]) and the oracle MLE. For $m \leq 6$, the proposed rank-breaking performs as well as an oracle who knows the hidden ranking among the top $m$ items. Jester dataset consists of continuous ratings between $-10$ to $+10$ of 100 jokes on sets of size $\kappa$, $36 \leq \kappa \leq 100$, by $24,983$ users. We convert ratings into full rankings. The ground truth $\theta^*$ is computed similarly. For $m \in \{2, 3, 4, 5\}$, we convert each full ranking into a poset that has $\ell = \lfloor \kappa/m \rfloor$ partitions of size $m$, by removing known relative ordering from each partition. Figure 4 compares the three algorithms using all samples (middle panel), and by varying the sample size (right panel) for fixed $m = 4$. All figures are averaged over 50 instances.

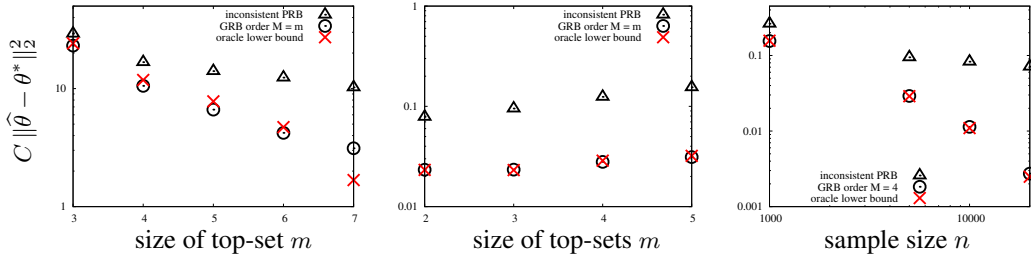

Figure 4: Generalized rank-breaking improves over pairwise RB and is close to oracle MLE.

## Acknowledgements

This work is supported by NSF SaTC award CNS-1527754, and NSF CISE award CCF-1553452.

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
