[Supplementary Material · rankbreaking_supplementary.pdf]

# Supplementary materials for "Computational and Statistical Tradeoffs in Learning to Rank"

## A  Proof of Remark 2.1

Recall that $\mathbb{P}_\theta(B(e) \prec T(e))$ is the probability that an agent ranks the collection of items $T(e)$ above $B(e)$ when offered $S = B(e) \cup T(e)$. We want to show that $\mathbb{P}_\theta(B(e) \prec T(e))$ is log-concave under the PL model. We prove a slightly general result which works for a family of RUMs in the location family. Random Utility Models (RUM) are defined as a probabilistic model where there is a real-valued utility parameter $\theta_i$ associated with each items $i \in S$, and an agent independently samples random utilities $\{U_i\}_{i \in S}$ for each item $i$ with conditional distribution $\mu_i(\cdot|\theta_i)$. Then the ranking is obtained by sorting the items in decreasing order as per the observed random utilities $U_i$'s. *Location family* is a subset of RUMs where the shapes of $\mu_i$'s are fixed and the only parameters are the means of the distributions. For location family, the noisy utilities can be written as $U_i = \theta_i + Z_i$ for i.i.d. random variable $Z_i$'s. In particular, it is PL model when $Z_i$'s follow the independent standard Gumbel distribution. We will show that for the location family if the probability density function for each $Z_i$'s is log-concave then $\log \mathbb{P}_\theta(B(e) \prec T(e))$ is concave. The desired claim follows as the pdf of standard Gumbel distribution is log-concave. We use the following Theorem from [20]. A similar technique was used to prove concavity when $|T(e)| = 1$ in [5].

**Lemma A.1** (Theorem 9 in [20]). *Suppose $g(\theta, Z)$ is a concave function in $\mathbb{R}^{2r}$, where $\theta \in \mathbb{R}^r$ is fixed and $Z$ is a $r-$component random vector whose probability distribution is logarithmic concave in $\mathbb{R}^r$, then the function*

$$h(\theta) = \mathbb{P}[g(\theta, Z) \geq 0], \qquad \text{for } \theta \in \mathbb{R}^r \tag{14}$$

*is logarithmic concave on $\mathbb{R}^r$.*

To apply the above lemma to get our result, let $r = |S|$, $g(\theta, Z) = \min_{i \in T(e)}\{\theta_i + Z_i\} - \max_{i' \in B(e)}\{\theta_{i'} + Z_{i'}\}$, and observe that $\mathbb{P}_\theta(B(e) \prec T(e)) = \mathbb{P}(g(\theta, Z) \geq 0)$ and $g(\theta, Z)$ is concave.

## B  Proof of Remark 2.2

Define event $E(e) \equiv \{T(e) \cup B(e)$ items are ranked in bottom $r$ positions when the offer set is $[d]\}$. Define $\mathbb{P}_{\theta,[d]}(B(e) \prec T(e)|E(e))$ be the conditional probability of $T(e)$ items being ranked higher than $B(e)$ items when the offer set is $[d]$, conditioned on the event $E(e)$. Observe that $\mathbb{P}_{\theta,[d]}(B(e) \prec T(e)|E(e))$ is the probability of observing the event $B(e) \prec T(e)$ under the proposed rank-breaking. First we show that $\mathbb{P}_\theta(e) = \mathbb{P}_{\theta,[d]}(B(e) \prec T(e)|E(e))$, where $\mathbb{P}_\theta(e)$ is the probability that $T(e) \prec B(e)$ when the offer set is $\{T(e) \cup B(e)\}$ as defined in (2). This follows from the fact that under PL model for any disjoint set of items $\{\mathcal{C}_i\}_{i \in [\ell]}$ such that $\cup_{i=1}^\ell \mathcal{C}_i = [d]$,

$$\mathbb{P}\big(\mathcal{C}_\ell \prec \mathcal{C}_{\ell-1} \prec \cdots \prec \mathcal{C}_1\big) = \mathbb{P}\big(\mathcal{C}_\ell \prec \mathcal{C}_{\ell-1}\big)\mathbb{P}\big(\{\mathcal{C}_\ell, \mathcal{C}_{\ell-1}\} \prec \mathcal{C}_{\ell-2}\big)\cdots \mathbb{P}\big(\{\mathcal{C}_\ell, \mathcal{C}_{\ell-1}, \cdots, \mathcal{C}_2\} \prec \mathcal{C}_1\big),$$
$$\tag{15}$$

where $\mathbb{P}(\mathcal{C}_{i_1} \prec \mathcal{C}_{i_2})$ is the probability that $\mathcal{C}_{i_2}$ items are ranked higher than $\mathcal{C}_{i_1}$ items when the offer set is $S = \{\mathcal{C}_{i_1} \cup \mathcal{C}_{i_2}\}$. Under the given sampling scenario, the comparison graph $\mathcal{H}([d], E)$ as defined in section 3 is connected and hence the estimate $\widehat{\theta}$, (3) is unique. Therefore, it follows that maximum likelihood estimate $\widehat{\theta}$ is consistent. Further, for a general sampling scenario, Theorem 4.1 proves that the estimator is consistent as the error goes to zero in the limit as $n$ increases.

## C  Proof of Theorem 4.1

We define few additional notations. $p \equiv (1/n)\sum_{j=1}^n p_j$. $V(e_{j,a}) \equiv T(e_{j,a}) \cup B(e_{j,a})$ for all $j \in [n]$ and $a \in [\ell_j]$. Note that by definition of rank-breaking edge $e_{j,a}$, $V(e_{j,a})$ is a random set of items that are ranked in bottom $r_{j,a}$ positions in a set of $S_j$ items by the user $j$.

The proof sketch is inspired from [15]. The main difference and technical challenge is in showing the strict concavity of $\mathcal{L}_{\mathrm{RB}}(\theta)$ when restricted to $\Omega_b$. We want to prove an upper bound on $\Delta = \widehat{\theta} - \theta^*$,

where $\widehat{\theta}$ is the sample dependent solution of the optimization (3) and $\theta^*$ is the true utility parameter from which the samples are drawn. Since $\widehat{\theta}, \theta^* \in \Omega_b$, it follows that $\Delta\mathbf{1} = 0$. Since $\widehat{\theta}$ is the maximizer of $\mathcal{L}_{\mathrm{RB}}(\theta)$, we have the following inequality,

$$\mathcal{L}_{\mathrm{RB}}(\widehat{\theta}) - \mathcal{L}_{\mathrm{RB}}(\theta^*) - \langle \nabla\mathcal{L}_{\mathrm{RB}}(\theta^*), \Delta \rangle \geq -\langle \nabla\mathcal{L}_{\mathrm{RB}}(\theta^*), \Delta \rangle \geq -\|\nabla\mathcal{L}_{\mathrm{RB}}(\theta^*)\|_2 \|\Delta\|_2, \quad (16)$$

where the last inequality uses the Cauchy-Schwartz inequality. By the mean value theorem, there exists a $\theta = c\widehat{\theta} + (1-c)\theta^*$ for some $c \in [0,1]$ such that $\theta \in \Omega_b$ and

$$\mathcal{L}_{\mathrm{RB}}(\widehat{\theta}) - \mathcal{L}_{\mathrm{RB}}(\theta^*) - \langle \nabla\mathcal{L}_{\mathrm{RB}}(\theta^*), \Delta \rangle = \frac{1}{2}\Delta^\top H(\theta)\Delta \leq -\frac{1}{2}\lambda_2(-H(\theta))\|\Delta\|_2^2, \quad (17)$$

where $\lambda_2(-H(\theta))$ is the second smallest eigen value of $-H(\theta)$. We will show in Lemma C.3 that $-H(\theta)$ is positive semi definite with one eigen value at zero with a corresponding eigen vector $\mathbf{1} = [1, \ldots, 1]^\top$. The last inequality follows since $\Delta^\top \mathbf{1} = 0$. Combining Equations (16) and (17),

$$\|\Delta\|_2 \leq \frac{2\|\nabla\mathcal{L}_{\mathrm{RB}}(\theta^*)\|_2}{\lambda_2(-H(\theta))}, \quad (18)$$

where we used the fact that $\lambda_2(-H(\theta)) > 0$ from Lemma C.3. The following technical lemmas prove that the norm of the gradient is upper bounded by $\gamma_2^{-1/2}e^b\sqrt{6np\log d}$ with high probability and the second smallest eigen value is lower bounded by $(1/8)\,e^{-6b}\alpha\gamma_1\gamma_2\gamma_3(np/(d-1))$. This finishes the proof of Theorem 4.1.

The (random) gradient of the log likelihood in (3) can be written as the following, where the randomness is in which items ended up in the top set $T(e_{j,a})$ and the bottom set $B(e_{j,a})$:

$$\nabla_i\mathcal{L}_{\mathrm{RB}}(\theta) = \sum_{j=1}^{n}\sum_{a=1}^{\ell_j}\sum_{\substack{\mathcal{C}\subseteq S_j, \\ |\mathcal{C}|=r_{j,a}-1}} \mathbb{I}\{V(e_{j,a}) = \{\mathcal{C}, i\}\}\frac{\partial \log \mathbb{P}_\theta(e_{j,a})}{\partial \theta_i}. \quad (19)$$

Note that we are intentionally decomposing each summand as a summation over all $\mathcal{C}$ of size $r_{j,a}-1$, such that we can separate the analysis of the expectation in the following lemma. The random variable $\mathbb{I}\{\{\mathcal{C}, i\} = V(e_{j,a})\}$ indicates that we only include one term for any given instance of the sample. Note that the event $\mathbb{I}\{\{\mathcal{C}, i\} = V(e_{j,a})\}$ is equivalent to the event that the $\{\mathcal{C}, i\}$ items are ranked in bottom $r_{j,a}$ positions in the set $S_j$, that is $V(e_{j,a})$ items are ranked in bottom $r_{j,a}$ positions in the set $S_j$.

**Lemma C.1.** *If the $j$-th poset is drawn from the PL model with weights $\theta^*$ then for any given $\mathcal{C}' \subseteq S_j$ with $|\mathcal{C}'| = r_{j,a}$,*

$$\mathbb{E}\left[\mathbb{I}\{\mathcal{C}' = V(e_{j,a})\}\frac{\partial \log \mathbb{P}_{\theta^*}(e_{j,a})}{\partial \theta_i^*}\Big|\{e_{j,a'}\}_{a'<a}\right] = 0. \quad (20)$$

First, this lemma implies that $\mathbb{E}\left[\mathbb{I}\{\mathcal{C}' = V(e_{j,a})\}\frac{\partial \log \mathbb{P}_{\theta^*}(e_{j,a})}{\partial \theta_i^*}\right] = 0$. Secondly, the above lemma allows us to construct a vector-valued martingale and apply a generalization of Azuma-Hoeffding's tail bound on the norm to prove the following concentration of measure. This proves the desired bound on the gradient.

**Lemma C.2.** *If $n$ posets are independently drawn over $d$ items from the PL model with weights $\theta^*$ then with probability at least $1 - 2e^3d^{-3}$,*

$$\|\nabla\mathcal{L}_{\mathrm{RB}}(\theta^*)\| \leq \gamma_2^{-1/2}e^b\sqrt{6np\log d}, \quad (21)$$

*where $\gamma_2$ depend on the choice of the rank-breaking and are defined in Section 3.*

We will prove in (29) that the Hessian matrix $H(\theta) \in \mathcal{S}^d$ with $H_{ii'}(\theta) = \frac{\partial^2 \mathcal{L}_{\mathrm{RB}}(\theta)}{\partial\theta_i\partial\theta_{i'}}$ can be expressed as

$$-H(\theta) = \sum_{j=1}^{n}\sum_{a=1}^{\ell_j}\sum_{i<i'\in S_j}\mathbb{I}\{(i,i')\subseteq V(e_{j,a})\}\left(\frac{\partial^2 \log \mathbb{P}_\theta(e_{j,a})}{\partial\theta_i\partial\theta_{i'}}(e_i - e_{i'})(e_i - e_{i'})^\top\right). \quad (22)$$

It is easy to see that $H(\theta)\mathbf{1} = 0$. The following lemma proves a lower bound on the second smallest eigenvalue $\lambda_2(-H(\theta))$ in terms of re-scaled spectral gap $\alpha$ of the comparison graph $\mathcal{H}$ defined in Section 3.

**Lemma C.3.** *Under the hypothesis of Theorem 4.1, if the assumptions in Equation* (9) *are satisfied then with probability at least* $1 - d^{-3}$, *the following holds for any* $\theta \in \Omega_b$:

$$\lambda_2(-H(\theta)) \geq \frac{e^{-6b}\alpha\gamma_1\gamma_2\gamma_3}{8} \frac{np}{(d-1)}, \tag{23}$$

*and* $\lambda_1(-H(\theta)) = 0$ *with corresponding eigen vector* $\mathbf{1}$.

This finishes the proof of the desired claim.

## C.1 Proof of Lemma C.1

Recall that $e_{j,a}$ is a random event where randomness is in which items ended up in the top-set $T(e_{j,a})$ and the bottom-set $B(e_{j,a})$, and $\mathbb{P}_{\theta^*}(e_{j,a}) = \mathbb{P}_{\theta^*}[B(e_{j,a}) \prec T(e_{j,a})]$ that is the probability of observing $B(e_{j,a}) \prec T(e_{j,a})$ when the offer set is $B(e_{j,a}) \cup T(e_{j,a})$ as defined in (2). Define, $\mathbb{P}_{\theta^*,S_j}[e_{j,a}|V(e_{j,a}) = \mathcal{C}']$ to be the conditional probability of observing $B(e_{j,a}) \prec T(e_{j,a})$, when the offer set is $S_j$, conditioned on the event that $V(e_{j,a}) = \mathcal{C}'$. Note that we have put subscript $S_j$ in $\mathbb{P}_{\theta^*}$ to specify that the offer set is $S_j$. Observe that for any set $\mathcal{C}' \subseteq S_j$, the event $\{\mathcal{C}' = V(e_{j,a})\}$ is equivalent to $\mathcal{C}'$ items being ranked in bottom $r_{j,a}$ positions when the offer set is $S_j$. In other words, it is conditioned on the event that the subset $V(e_{j,a})$ items are ranked in bottom $r_{j,a}$ positions when the offer set is $S_j$. It is easy to check that under PL model

$$\mathbb{P}_{\theta^*,S_j}[e_{j,a}|V(e_{j,a}) = \mathcal{C}'] = \mathbb{P}_{\theta^*}[e_{j,a}],$$

(see Remark 2.2). Also, by conditioning on any outcome of $\{e_{j,a'}\}_{a'<a}$ it can be checked that

$$\mathbb{P}_{\theta^*,S_j}[e_{j,a}|V(e_{j,a}) = \mathcal{C}', \{e_{j,a'}\}_{a'<a}] = \mathbb{P}_{\theta^*,S_j}[e_{j,a}|V(e_{j,a}) = \mathcal{C}'].$$

Therefore, we have

$$\mathbb{E}\left[\frac{\partial \log \mathbb{P}_{\theta^*}[e_{j,a}]}{\partial \theta_i^*}\bigg| V(e_{j,a}) = \mathcal{C}', \{e_{j,a'}\}_{a'<a}\right]$$

$$= \mathbb{E}\left[\frac{\partial \log \mathbb{P}_{\theta^*,S_j}[e_{j,a}|V(e_{j,a}) = \mathcal{C}', \{e_{j,a'}\}_{a'<a}]}{\partial \theta_i^*}\bigg| V(e_{j,a}) = \mathcal{C}', \{e_{j,a'}\}_{a'<a}\right]$$

$$= \sum_{\substack{e_{j,a}:V(e_{j,a})=\mathcal{C}' \\ \{e_{j,a'}\}_{a'<a}}} \mathbb{P}_{\theta^*,S_j}\left[e_{j,a}\big|V(e_{j,a}) = \mathcal{C}', \{e_{j,a'}\}_{a'<a}\right] \frac{\partial}{\partial \theta_i^*} \log \mathbb{P}_{\theta^*,S_j}\left[e_{j,a}\big|V(e_{j,a}) = \mathcal{C}', \{e_{j,a'}\}_{a'<a}\right]$$

$$= \frac{\partial}{\partial \theta_i^*} \sum_{e_{j,a}:V(e_{j,a})=\mathcal{C}'} \mathbb{P}_{\theta^*,S_j}\left[e_{j,a}\big|V(e_{j,a}) = \mathcal{C}'\right] = \frac{\partial}{\partial \theta_i^*}1 = 0,$$

where we used $\{e_{j,a} : V(e_{j,a}) = \mathcal{C}'\} = \{e_{j,a} : V(e_{j,a}) = \mathcal{C}', \{e_{j,a'}\}_{a'<a}\}$ which follows from the definition of rank-breaking edges $e_{j,a}$. This proves the desired claim.

## C.2 Proof of Lemma C.2

We view $\nabla\mathcal{L}_{\mathrm{RB}}(\theta^*)$ as the final value of a discrete time vector-valued martingale with values in $\mathbb{R}^d$. Define $\nabla\mathcal{L}_{\mathrm{RB}}^{(e_{j,a})} \in \mathbb{R}^d$ as the gradient vector arising out of each rank-breaking edge $\{e_{j,a}\}_{j\in[n],a\in[\ell_j]}$ as

$$\nabla_i\mathcal{L}_{\mathrm{RB}}^{(e_{j,a})}(\theta^*) \equiv \sum_{\mathcal{C}\subseteq S_j} \mathbb{I}\{V(e_{j,a}) = \{\mathcal{C},i\}\}\nabla_i \log \mathbb{P}_{\theta^*}(e_{j,a}), \tag{24}$$

such that $\nabla\mathcal{L}_{\mathrm{RB}}(\theta^*) = \sum_{j\in[n]}\sum_{a\in[\ell_j]} \nabla\mathcal{L}_{\mathrm{RB}}^{(e_{j,a})}$. We take $\nabla\mathcal{L}_{\mathrm{RB}}^{(e_{j,a})}$ as the incremental random vector in a martingale of $\sum_{j=1}^{n}\ell_j$ time steps. Let $H_{j,a}$ denote (the sigma algebra of) the history up to $e_{j,a}$ and define a sequence of random vectors in $\mathbb{R}^d$:

$$Z_{j,a} \equiv \mathbb{E}[\nabla\mathcal{L}_{\mathrm{RB}}^{(e_{j,a})}(\theta^*)|H_{j,a}],$$

with the convention that $Z_{1,1} = \mathbb{E}[\nabla\mathcal{L}_{\mathrm{RB}}^{(e_{j,a})}(\theta^*)] = 0$ as proved in Lemma C.1. It also follows from Lemma C.1 that $\mathbb{E}[Z_{j,a+1}|Z_{j,a}] = Z_{j,a}$ for $a < \ell_j$. Also, from the independence of samples, it

follows that $\mathbb{E}[Z_{j+1,1}|Z_{j,\ell_j}] = Z_{j,\ell_j}$. Applying a generalized version of the vector Azuma-Hoeffding inequality which readily follows from [Theorem 1.8, [13]], we have

$$\mathbb{P}\big[\,\|\nabla\mathcal{L}_{\mathrm{RB}}(\theta^*)\| \geq \delta\,\big] \quad \leq \quad 2e^3 \exp\left(-\frac{\delta^2}{\sum_{j=1}^n \sum_{a=1}^{\ell_j} m_{j,a}2\gamma_2^{-1}e^{2b}}\right), \tag{25}$$

where we used $\|\nabla\mathcal{L}_{\mathrm{RB}}^{(e_{j,a})}\|^2 \leq m_{j,a}2\gamma_2^{-1}e^{2b}$. Choosing $\delta = \gamma_2^{-1}e^b\sqrt{6np\log d}$ gives the desired bound.

Now we are left to show that $\|\nabla\mathcal{L}_{\mathrm{RB}}^{(e_{j,a})}\|^2 \leq 2m_{j,a}\gamma_2^{-1}e^{2b}$ for any $\theta \in \Omega_b$. Recall that $\sigma \in \Lambda_{T(e_{j,a})}$ is the set of all full rankings over $T(e_{j,a})$ items. In rest of the proof, with a slight abuse of notations, we extend each of these ranking $\sigma$ over $T(e_{j,a}) \cup B(e_{j,a})$ items in the following way. Consider any full ranking $\tilde{\sigma}$ over $B(e_{j,a})$ items. Then for each $\sigma \in \Lambda_{T(e_{j,a})}$, the extension is such that $\sigma(|T(e_{j,a})| + c) = \tilde{\sigma}(c)$ for $1 \leq c \leq |B(e_{j,a})|$. The choice of ranking $\tilde{\sigma}$ will have no impact on any of the following mathematical expressions. From the definition of $\mathbb{P}_\theta(e_{j,a})$ (2), we have, for any $i \in V(e_{j,a})$,

$$\frac{\partial\mathbb{P}_\theta(e_{j,a})}{\partial\theta_i} = \mathbb{I}\{i \in T(e_{j,a})\}\mathbb{P}_\theta(e_{j,a}) \tag{26}$$

$$- \underbrace{\sum_{\sigma\in\Lambda_{T(e_{j,a})}} \underbrace{\frac{\exp\left(\sum_{c=1}^{m_{j,a}}\theta_{\sigma(c)}\right)}{\prod_{u=1}^{m_{j,a}}\left(\sum_{c'=u}^{r_{j,a}}\exp\left(\theta_{\sigma(c')}\right)\right)}}_{\equiv A_\sigma} \underbrace{\left(\sum_{u'=1}^{m_{j,a}} \frac{\mathbb{I}\{\sigma^{-1}(i)\geq u'\}\exp(\theta_i)}{\sum_{c'=u'}^{r_{j,a}}\exp\left(\theta_{\sigma(c')}\right)}\right)}_{\equiv B_{\sigma,i}}}_{\equiv E_i}.$$

Note that $A_\sigma, B_{\sigma,i}$ and $E_i$ depend on $e_{j,a}$. Observe that for any $1 \leq u' \leq m_{j,a}$ and any $\sigma \in \Lambda_{T(e_{j,a})}$,

$$\sum_{i\in V(e_{j,a})} \mathbb{I}\{\sigma^{-1}(i)\geq u'\}\exp(\theta_i) = \sum_{c'=u'}^{r_{j,a}}\exp\left(\theta_{\sigma(c')}\right). \tag{27}$$

Therefore, $\sum_{i\in V(e_{j,a})} B_{\sigma,i} = m_{j,a}$. It follows that

$$\sum_{i\in V(e_{j,a})} E_i = \sum_{\sigma\in\Lambda_{T(e_{j,a})}} A_\sigma\left(\sum_{i\in V(e_{j,a})} B_{\sigma,i}\right) = m_{j,a}\sum_{\sigma\in\Lambda_{T(e_{j,a})}} A_\sigma = m_{j,a}\mathbb{P}_\theta(e_{j,a}), \tag{28}$$

where the last equality follows from the definition of $\mathbb{P}_\theta(e_{j,a})$ (3). Also, since for any $i, i'$, $e^{(\theta_i-\theta_{i'})} \leq e^{2b}$; for any $i$, $B_{\sigma,i} \leq e^{2b}\sum_{k=r_{j,a}-m_{j,a}+1}^{r_{j,a}}(1/k) \leq e^{2b}(1 + \log(r_{j,a}/(r_{j,a} - m_{j,a} + 1))) \leq \gamma_2^{-1}e^{2b}$, where the last inequality follows from the definition of $\gamma_2$ (7) and the fact that $x \leq \sqrt{1+\log x}$ for all $x \geq 1$. Therefore, $E_i \leq \gamma_2^{-1}e^{2b}\sum_{\sigma\in\Lambda_{T(e_{j,a})}} A_\sigma = \gamma_2^{-1}e^{2b}\mathbb{P}_\theta(e_{j,a})$. We have $\partial\log\mathbb{P}_\theta(e_{j,a})/\partial\theta_i = (1/\mathbb{P}_\theta(e_{j,a}))\partial\mathbb{P}_\theta(e_{j,a})/\partial\theta_i = \mathbb{I}\{i \in T(e_{j,a})\} - E_i/\mathbb{P}_\theta(e_{j,a})$. Since $|T(e_{j,a})| = m_{j,a}$, $\|\nabla\mathcal{L}_{\mathrm{RB}}^{(e_{j,a})}\|^2 \leq m_{j,a} + \sum_{i\in V(e_{j,a})}(E_i/\mathbb{P}_\theta(e_{j,a}))^2 \leq 2m_{j,a}\gamma_2^{-1}e^{2b}$, where we used (28) and the fact that $\gamma_2^{-1} \geq 1$.

### C.2.1 Proof of Lemma C.3

First, we prove (22). For brevity, remove $\{j, a\}$ from $\mathbb{P}_\theta(e_{j,a})$. From Equations (26) and (28), and $|T(e_{j,a})| = m_{j,a}$, we have $\sum_{i\in V(e_{j,a})} \frac{\partial}{\partial\theta_i}\mathbb{P}_\theta(e) = m_{j,a}\mathbb{P}_\theta(e) - m_{j,a}\mathbb{P}_\theta(e) = 0$. It follows that

$$\sum_{i\in V(e_{j,a})}\left(\frac{\partial^2\log\mathbb{P}_\theta(e)}{\partial\theta_{i'}\partial\theta_i}\right) =$$

$$\frac{1}{\mathbb{P}_\theta(e)}\frac{\partial}{\partial\theta_{i'}}\left(\sum_{i\in V(e_{j,a})}\left(\frac{\partial\mathbb{P}_\theta(e)}{\partial\theta_i}\right)\right) - \frac{1}{(\mathbb{P}_\theta(e))^2}\frac{\partial\mathbb{P}_\theta(e)}{\partial\theta_{i'}}\left(\sum_{i\in V(e_{j,a})}\left(\frac{\partial\mathbb{P}_\theta(e)}{\partial\theta_i}\right)\right) = 0. \tag{29}$$

Since by definition $\mathcal{L}_{\mathrm{RB}}(\theta) = \sum_{j=1}^n \sum_{a=1}^{\ell_j}\log\mathbb{P}_\theta(e_{j,a})$, and $H_{ii'}(\theta) = \frac{\partial^2\mathcal{L}_{\mathrm{RB}}(\theta)}{\partial\theta_i\partial\theta_{i'}}$ which is a symmetric matrix, Equation (29) implies that it can be expressed as given in Equation (22). It follows that all-ones is an eigenvector of $H(-\theta)$ with the corresponding eigenvalue being zero.

To get a lower bound on $\lambda_2(-H(\theta))$, we apply Weyl's inequality

$$\lambda_2(-H(\theta)) \geq \lambda_2(\mathbb{E}[-H(\theta)]) - \|H(\theta) - \mathbb{E}[H(\theta)]\|. \tag{30}$$

We will show in (33) that $\lambda_2(\mathbb{E}[-H(\theta)]) \geq e^{-6b}\alpha\gamma_1\gamma_2\gamma_3(np/(4(d-1)))$ and in (50) that $\|H(\theta) - \mathbb{E}[H(\theta)]\| \leq 16e^{4b}\nu\sqrt{\frac{p_{\max}}{\kappa_{\min}}\frac{np}{\beta(d-1)}\log d}$. Putting these together,

$$\lambda_2(-H(\theta)) \geq e^{-6b}\alpha\gamma_1\gamma_2\gamma_3\frac{np}{4(d-1)} - 16e^{4b}\nu\sqrt{\frac{p_{\max}}{\kappa_{\min}}\frac{np}{\beta(d-1)}\log d} \tag{31}$$

$$\geq \frac{e^{-6b}\alpha\gamma_1\gamma_2\gamma_3}{8}\frac{np}{(d-1)}, \tag{32}$$

where the last inequality follows from the assumption on $n\kappa_{\min}$ given in (9).

To prove a lower bound on $\lambda_2(\mathbb{E}[-H(\theta)])$, we claim that for $\theta \in \Omega_b$,

$$\mathbb{E}[-H(\theta)] \succeq e^{-6b}\gamma_1\gamma_2\gamma_3\sum_{j=1}^{n}\frac{p_j}{4\kappa_j(\kappa_j-1)}\sum_{i<i'\in S_j}(e_i-e_{i'})(e_i-e_{i'})^\top \tag{33}$$

$$= \frac{e^{-6b}\gamma_1\gamma_2\gamma_3}{4}L,$$

where $L \in \mathcal{S}^d$ is defined in (5). Using $\lambda_2(L) = np\alpha/(d-1)$ from (6), we have $\lambda_2(-H(\theta)) \geq e^{-6b}\alpha\gamma_1\gamma_2\gamma_3(np/(4(d-1)))$. To prove (33), notice that

$$\mathbb{E}[-H(\theta)_{ii'}] = \mathbb{E}\left[\sum_{j\in[n]}\sum_{a\in[\ell_j]}\mathbb{I}\{(i,i')\subseteq V(e_{j,a})\}\frac{\partial^2\log\mathbb{P}_\theta(e_{j,a})}{\partial\theta_i\partial\theta_{i'}}\right], \tag{34}$$

when $i \neq i'$. We will show that for any $i \neq i' \in V(e_{j,a})$,

$$\frac{\partial^2\log\mathbb{P}_\theta(e_{j,a})}{\partial\theta_i\partial\theta_{i'}} \geq \begin{cases} \dfrac{e^{-2b}m_{j,a}}{r_{j,a}^2} & \text{if } i,i'\in B(e_{j,a}) \\[2ex] -\dfrac{e^{4b}m_{j,a}^2}{(r_{j,a}-m_{j,a}+1)^2} & \text{otherwise}. \end{cases} \tag{35}$$

We need to bound the probability of two items appearing in the bottom-set $B(e_{j,a})$ and in the top-set $T(e_{j,a})$.

**Lemma C.4.** *Consider a ranking $\sigma$ over a set $S \subseteq [d]$ such that $|S| = \kappa$. For any two items $i, i' \in S$, $\theta \in \Omega_b$, and $1 \leq \ell, \ell_1, \ell_2 \leq \kappa - 1$,*

$$\mathbb{P}_\theta[\sigma^{-1}(i), \sigma^{-1}(i') > \ell] \geq \frac{e^{-4b}(\kappa-\ell)(\kappa-\ell-1)}{\kappa(\kappa-1)}\left(1-\frac{\ell}{\kappa}\right)^{2e^{2b}-2}, \tag{36}$$

$$\mathbb{P}_\theta[\sigma^{-1}(i) = \ell] \leq \frac{e^{6b}}{\kappa-\ell}, \tag{37}$$

$$\mathbb{P}_\theta[\sigma^{-1}(i) = \ell_1, \sigma^{-1}(i') = \ell_2] \leq \frac{e^{10b}}{(\kappa-\ell_1-1)(\kappa-\ell_2)}. \tag{38}$$

*where the probability $\mathbb{P}_\theta$ is with respect to the sampled ranking resulting from PL weights $\theta \in \Omega_b$.*

Substituting $\ell = \kappa_j - r_{j,a} + m_{j,a}$ in (36), and $\ell, \ell_1, \ell_2 \leq \kappa_j - r_{j,a} + m_{j,a}$ in (37) and (38), we have,

$$\mathbb{P}_\theta[(i,i')\subseteq B(e_{j,a})] \geq \frac{e^{-4b}(r_{j,a}-m_{j,a})^2}{4\kappa_j(\kappa_j-1)}\left(\frac{r_{j,a}-m_{j,a}}{\kappa_j}\right)^{2e^{2b}-2}, \tag{39}$$

$$\mathbb{P}_\theta[i\in T(e_{j,a}), i'\in B(e_{j,a})] \leq m_{j,a}\max_{\ell\in[\kappa_j-r_{j,a}+m_{j,a}]}\mathbb{P}(\sigma^{-1}(i)=\ell)$$

$$\leq \frac{e^{6b}m_{j,a}}{r_{j,a}-m_{j,a}}, \tag{40}$$

$$\mathbb{P}_\theta[(i,i')\subseteq T(e_{j,a})] \leq m_{j,a}^2\max_{\ell_1,\ell_2\in[\kappa_j-r_{j,a}+m_{j,a}]}\mathbb{P}(\sigma^{-1}(i)=\ell_1,\sigma^{-1}(i')=\ell_2)$$

$$\leq \frac{e^{10b}m_{j,a}^2}{2(r_{j,a}-m_{j,a}-1)(r_{j,a}-m_{j,a})}, \tag{41}$$

where (39) uses $r_{j,a} - m_{j,a} - 1 \geq (r_{j,a} - m_{j,a})/4$, (40) uses $\mathbb{P}_\theta[i \in T(e_{j,a}), i' \in B(e_{j,a})] \leq \mathbb{P}_\theta[i \in T(e_{j,a})]$, and (40)-(41) uses counting on the possible choices. The bound in (41) is smaller than the one in (40) as per our assumption that $\gamma_3 > 0$.

Using Equations (34)-(35) and (39)-(41), and the definitions of $\gamma_1, \gamma_2, \gamma_3$ from Section 3, we get $\mathbb{E}[-H(\theta)_{ii'}] \geq$

$$\sum_{j \in [n]} \sum_{a \in [\ell_j]} \Big\{ \underbrace{\left( \frac{r_{j,a} - m_{j,a}}{\kappa_j} \right)^{2e^{2b}-2}}_{\geq \gamma_1} \underbrace{\left( \frac{r_{j,a} - m_{j,a}}{r_{j,a}} \right)^2 \frac{e^{-6b} m_{j,a}}{4\kappa_j(\kappa_j - 1)}}_{\geq \gamma_2} - \frac{e^{6b} m_{j,a}}{r_{j,a} - m_{j,a}} \frac{e^{4b} m_{j,a}^2}{(r_{j,a} - m_{j,a} + 1)^2} \Big\}$$

$$\geq \sum_{j,a} \frac{\gamma_1 \gamma_2 e^{-6b} m_{j,a}}{4\kappa_j(\kappa_j - 1)} \underbrace{\left( 1 - \frac{4e^{16b}}{\gamma_1} \frac{m_{j,a}^2 r_{j,a}^2 \kappa_j^2}{(r_{j,a} - m_{j,a})^5} \right)}_{\geq \gamma_3} . \tag{42}$$

This combined with (22) proves the desired claim (33). Further, in Appendix E, we show that if $m_{j,a} \leq 3$ for all $\{j, a\}$ then $\frac{\partial^2 \log \mathbb{P}_\theta(e_{j,a})}{\partial\theta_i \partial\theta_{i'}}$ is non-negative even for $i \neq i' \in T(e_{j,a})$, and $i \in T(e_{j,a}), i' \in B(e_{j,a})$ as opposed to a negative lower-bound given in (35). Therefore, bound on $\mathbb{E}[-H(\theta)]$ in (33) can be tightened by a factor of $\gamma_3$.

To prove claim (35), define the following for $\sigma \in \Lambda_{T(e_{j,a})}$,

$$A_\sigma \equiv \frac{\exp\left(\sum_{c=1}^{m_{j,a}} \theta_{\sigma(c)}\right)}{\prod_{u=1}^{m_{j,a}} \left(\sum_{c'=u}^{r_{j,a}} \exp\left(\theta_{\sigma(c')}\right)\right)}, B_\sigma \equiv \sum_{u'=1}^{m_{j,a}} \frac{1}{\sum_{c'=u'}^{r_{j,a}} \exp\left(\theta_{\sigma(c')}\right)},$$

$$B_{\sigma,i} \equiv \sum_{u'=1}^{m_{j,a}} \frac{\mathbb{I}\{\sigma^{-1}(i) \geq u'\}}{\sum_{c'=u'}^{r_{j,a}} \exp\left(\theta_{\sigma(c')}\right)}, C_\sigma \equiv \sum_{u'=1}^{m_{j,a}} \frac{1}{\left(\sum_{c'=u'}^{r_{j,a}} \exp\left(\theta_{\sigma(c')}\right)\right)^2},$$

$$C_{\sigma,i} \equiv \sum_{u'=1}^{m_{j,a}} \frac{\mathbb{I}\{\sigma^{-1}(i) \geq u'\}}{\left(\sum_{c'=u'}^{r_{j,a}} \exp\left(\theta_{\sigma(c')}\right)\right)^2}, C_{\sigma,i,i'} \equiv \sum_{u'=1}^{m_{j,a}} \frac{\mathbb{I}\{\sigma^{-1}(i), \sigma^{-1}(i') \geq u'\}}{\left(\sum_{c'=u'}^{r_{j,a}} \exp\left(\theta_{\sigma(c')}\right)\right)^2} . \tag{43}$$

First, a few observations about the expression of $A_\sigma$. For any $\sigma \in \Lambda_{T(e_{j,a})}$ and any $i \in V(e_{j,a})$, $\theta_i$ is in the numerator if and only if $i \in T(e_{j,a})$, since in all the rankings that are consistent with the observation $e_{j,a}$, $T(e_{j,a})$ items are ranked in top $m_{j,a}$ positions. For any $\sigma \in \Lambda_{T(e_{j,a})}$ and any $i \in B(e_{j,a})$, $\theta_i$ is in all the product terms $\prod_{u=1}^{m_{j,a}}(\cdot)$ of the denominator, since in all the consistent rankings these items are ranked below $m_{j,a}$ position. For any $i \in T(e_{j,a})$, $\theta_i$ appears in product term corresponding to index $u$ if and only if item $i$ is ranked at position $u$ or lower than $u$ in the ranking $\sigma \in \Lambda_{T(e_{j,a})}$. Now, observe that $B_\sigma$ is defined such that the partial derivative of $A_\sigma$ with respect to any $i \in B(e_{j,a})$ is $-A_\sigma B_\sigma e^{\theta_i}$, and $B_{\sigma,i}$ is defined such that the partial derivative of $A_\sigma$ with respect to any $i \in T(e_{j,a})$ is $A_\sigma - A_\sigma B_{\sigma,i} e^{\theta_i}$. Further, observe that $-C_\sigma e^{\theta_i}$ is the partial derivative of $B_\sigma$ with respect to $i \in B(e_{j,a})$, $-C_{\sigma,i} e^{\theta_i}$ is the partial derivative of $B_{\sigma,i}$ with respect to $i \in T(e_{j,a})$, and $-C_{\sigma,i} e^{\theta_{i'}}$ is the partial derivative of $B_{\sigma,i}$ with respect to $i' \in B(e_{j,a})$. $-C_{\sigma,i,i'} e^{\theta_{i'}}$ is the partial derivative of $B_{\sigma,i}$ with respect to $i' \neq i \in T(e_{j,a})$.

For ease of notation, we omit subscript $(j, a)$ whenever it is clear from the context. Also, we use $\sum_\sigma$ to denote $\sum_{\sigma \in \Lambda_{T(e_{j,a})}}$. With the above defined notations, from (3), we have, $\mathbb{P}_\theta(e) = \sum_\sigma A_\sigma$.

With the above given observations for the notations in (43), first partial derivative of $\mathbb{P}_\theta(e)$ can be expressed as following:

$$\frac{\partial \mathbb{P}_\theta(e)}{\partial\theta_i} = \begin{cases} \sum_\sigma \left(A_\sigma - A_\sigma B_{\sigma,i} e^{\theta_i}\right) & \text{if } i \in T(e_{j,a}) \\ \sum_\sigma \left(-A_\sigma B_\sigma e^{\theta_i}\right) & \text{if } i \in B(e_{j,a}) . \end{cases} \tag{44}$$

It follows that for $i \neq i' \in V(e_{j,a})$,

$$\frac{\partial^2 \mathbb{P}_\theta(e)}{\partial\theta_i \partial\theta_{i'}}$$

$$= \begin{cases} \sum_\sigma \left((A_\sigma (B_\sigma)^2 + A_\sigma C_\sigma) e^{(\theta_i + \theta_{i'})}\right) & \text{if } i, i' \in B(e_{j,a}) \\ \sum_\sigma \left(A_\sigma - A_\sigma B_{\sigma,i'} e^{\theta_{i'}} + (A_\sigma B_{\sigma,i} B_{\sigma,i'} + A_\sigma C_{\sigma,i,i'}) e^{(\theta_i + \theta_{i'})} - A_\sigma B_{\sigma,i} e^{\theta_i}\right) & \text{if } i, i' \in T(e_{j,a}) \\ \sum_\sigma \left((A_\sigma B_\sigma B_{\sigma,i} + A_\sigma C_{\sigma,i}) e^{(\theta_i + \theta_{i'})} - A_\sigma B_\sigma e^{\theta_{i'}}\right) & \text{otherwise} . \end{cases} \tag{45}$$

Using $\frac{\partial^2 \log \mathbb{P}_\theta(e)}{\partial \theta_i \partial \theta_{i'}} = \frac{1}{\mathbb{P}_\theta(e)} \frac{\partial^2 \mathbb{P}_\theta(e)}{\partial \theta_i \partial \theta_{i'}} - \frac{1}{(\mathbb{P}_\theta(e))^2} \frac{\partial \mathbb{P}_\theta(e)}{\partial \theta_i} \frac{\partial \mathbb{P}_\theta(e)}{\partial \theta_{i'}}$, with above derived first and second derivatives, and after following some algebra, we have

$$
\frac{(\mathbb{P}_\theta(e))^2}{e^{(\theta_i + \theta_{i'})}} \frac{\partial^2 \log \mathbb{P}_\theta(e)}{\partial \theta_i \partial \theta_{i'}}
$$
$$
= \begin{cases}
(\sum_\sigma A_\sigma)(\sum_\sigma A_\sigma (B_\sigma)^2) - (\sum_\sigma A_\sigma B_\sigma)^2 + (\sum_\sigma A_\sigma)(\sum_\sigma A_\sigma C_\sigma) & \text{if } i, i' \in B(e_{j,a}) \\
(\sum_\sigma A_\sigma)(\sum_\sigma A_\sigma B_{\sigma,i} B_{\sigma,i'} + A_\sigma C_{\sigma,i,i'}) - (\sum_\sigma A_\sigma B_{\sigma,i})(\sum_\sigma A_\sigma B_{\sigma,i'}) & \text{if } i, i' \in T(e_{j,a}) \\
(\sum_\sigma A_\sigma)(\sum_\sigma A_\sigma B_\sigma B_{\sigma,i} + A_\sigma C_{\sigma,i}) - (\sum_\sigma A_\sigma B_\sigma)(\sum_\sigma A_\sigma B_{\sigma,i}) & \text{otherwise .}
\end{cases}
$$
$$(46)$$

Observe that from Cauchy-Schwartz inequality $(\sum_\sigma A_\sigma)(\sum_\sigma A_\sigma (B_\sigma)^2) - (\sum_\sigma A_\sigma B_\sigma)^2 \geq 0$. Also, we have $e^{(\theta_i + \theta_{i'})} C_\sigma \geq e^{-2b}(m/r^2)$ and $e^{\theta_i} B_{\sigma,i} \leq e^{\theta_i} B_\sigma \leq e^{2b}(m/(r - m + 1))$ for any $i \in V(e_{j,a})$. This proves the desired claim (35).

Next we need to upper bound deviation of $-H(\theta)$ from its expectation. From (46), we have, $\left| \frac{\partial^2 \log \mathbb{P}_\theta(e_{j,a})}{\partial \theta_i \partial \theta_{i'}} \right| \leq 3e^{4b} m_{j,a}^2 / (r_{j,a} - m_{j,a} + 1)^2 \leq 3e^{4b} \nu m_{j,a} / (\kappa_j(\kappa_j - 1))$, where the last inequality follows from the definition of $\nu$ (8). Therefore,

$$
-H(\theta) \preceq 3e^{4b}\nu \sum_{j=1}^{n} \sum_{a=1}^{\ell_j} \sum_{i < i' \in S_j} \mathbb{I}\{(i, i') \subseteq V(e_{j,a})\} \frac{m_{j,a}}{\kappa_j(\kappa_j - 1)} (e_i - e_{i'})(e_i - e_{i'})^\top \quad (47)
$$

$$
\preceq 3e^{4b}\nu \sum_{j=1}^{n} \sum_{i < i' \in S_j} \frac{\sum_{a=1}^{\ell_j} m_{j,a}}{\kappa_j(\kappa_j - 1)} (e_i - e_{i'})(e_i - e_{i'})^\top \equiv \sum_{j=1}^{n} y_j L_j , \quad (48)
$$

where $y_j = (3e^{4b}\nu p_j)/(\kappa_j(\kappa_j - 1))$ and $L_j = \sum_{i < i' \in S_j}(e_i - e_{i'})(e_i - e_{i'})^\top = \kappa_j \mathrm{diag}(e_{S_j}) - e_{S_j} e_{S_j}^\top$ for $e_{S_j} = \sum_{i \in S_j} e_i$. Observe that $\|y_j L_j\| \leq (3e^{4b}\nu p_{\max})/\kappa_{\min}$. Moreover, $L_j^2 \preceq \kappa_j L_j$, and it follows that

$$
\sum_{j=1}^{n} y_j^2 L_j^2 \preceq 9e^{8b}\nu^2 \sum_{j=1}^{n} \frac{p_j^2}{\kappa_j^2(\kappa_j - 1)^2} \kappa_j L_j \preceq \frac{9e^{8b}\nu^2 p_{\max}}{\kappa_{\min}} L , \quad (49)
$$

where we used the fact that $L = (p_j/(\kappa_j(\kappa_j - 1))) \sum_{j=1}^{n} L_j$, for $L$ defined in (5). Using $\lambda_d(L) = np/(\beta(d - 1))$ from (6), it follows that $\|\sum_{j=1}^{n} \mathbb{E}_\theta[y_j^2 Y_j^2]\| \leq \frac{9e^{8b}\nu^2 p_{\max}}{\kappa_{\min}} \frac{np}{\beta(d - 1)}$. By the matrix Bernstien inequality, with probability at least $1 - d^{-3}$,

$$
\|H(\theta) - \mathbb{E}[H(\theta)]\| \leq 12e^{4b}\nu \sqrt{\frac{p_{\max}}{\kappa_{\min}} \frac{np}{\beta(d - 1)} \log d} + \frac{8e^{4b}\nu p_{\max} \log d}{\kappa_{\min}}
$$

$$
\leq 16e^{4b}\nu \sqrt{\frac{p_{\max}}{\kappa_{\min}} \frac{np}{\beta(d - 1)} \log d} , \quad (50)
$$

where the last inequality follows from the assumption on $n\kappa_{\min}$ given in (9).

## C.3 Proof of Lemma C.4

**Claim** (36): Since providing a lower bound on $\mathbb{P}_\theta\left[\sigma^{-1}(i), \sigma^{-1}(i') > \ell\right]$ for arbitrary $\theta$ is challenging, we construct a new set of parameters $\{\widetilde{\theta}_j\}_{j \in [d]}$ from the original $\theta$. These new parameters are constructed such that it is both easy to compute the probability and also provides a lower bound on the original distribution. Define $\widetilde{\alpha}_{i,i',\ell,\theta}$ as

$$
\widetilde{\alpha}_{i,i',\ell,\theta} \equiv \max_{\ell' \in [\ell]} \max_{\substack{\Omega \subseteq S \setminus \{i,i'\} \\ : |\Omega| = \kappa - \ell'}} \left\{ \frac{\exp(\theta_i) + \exp(\theta_{i'})}{\left(\sum_{j \in \Omega} \exp(\theta_j)\right)/|\Omega|} \right\} , \quad (51)
$$

and $\alpha_{i,i',\ell,\theta} = \lceil \widetilde{\alpha}_{i,i',\ell,\theta} \rceil$. For ease of notation we remove the subscript from $\alpha$ and $\widetilde{\alpha}$. We denote the sum of the weights by $W \equiv \sum_{j \in S} \exp(\theta_j)$. We define a new set of parameters $\{\widetilde{\theta}_j\}_{j \in S}$:

$$
\widetilde{\theta}_j = \begin{cases} \log(\widetilde{\alpha}/2) & \text{for } j = i \text{ or } i' , \\ 0 & \text{otherwise .} \end{cases} \quad (52)
$$

Similarly define $\widetilde{W} \equiv \sum_{j \in S} \exp(\widetilde{\theta}_j) = \kappa - 2 + \widetilde{\alpha}$. We have,

$$
\begin{aligned}
&\mathbb{P}_\theta\left[\sigma^{-1}(i), \sigma^{-1}(i') > \ell\right] \\
=\ & \sum_{\substack{j_1 \in S \\ j_1 \neq i, i'}} \left(\frac{\exp(\theta_{j_1})}{W} \sum_{\substack{j_2 \in S \\ j_2 \neq i, i', j_1}} \left(\frac{\exp(\theta_{j_2})}{W - \exp(\theta_{j_1})} \cdots \left(\sum_{\substack{j_\ell \in S \\ j_\ell \neq i, i', \\ j_1, \cdots, j_{\ell-1}}} \frac{\exp(\theta_{j_\ell})}{W - \sum_{k=j_1}^{j_{\ell-1}} \exp(\theta_k)}\right) \cdots \right)\right) \\
=\ & \sum_{\substack{j_1 \in S \\ j_1 \neq i, i'}} \left(\frac{\exp(\theta_{j_1})}{W - \exp(\theta_{j_1})} \cdots \sum_{\substack{j_{\ell-1} \in S \\ j_{\ell-1} \neq i, i', \\ j_1, \cdots, j_{\ell-2}}} \left(\frac{\exp(\theta_{j_{\ell-1}})}{W - \sum_{k=j_1}^{j_{\ell-1}} \exp(\theta_k)} \sum_{\substack{j_\ell \in S \\ j_\ell \neq i, i', \\ j_1, \cdots, j_{\ell-1}}} \left(\frac{\exp(\theta_{j_\ell})}{W}\right) \cdots \right)\right)
\end{aligned}
$$
(53)

Consider the second-last summation term in the above equation and let $\Omega_\ell = S \setminus \{i, i', j_1, \ldots, j_{\ell-2}\}$. Observe that, $|\Omega_\ell| = \kappa - \ell$ and from equation (51), $\frac{\exp(\theta_i) + \exp(\theta_{i'})}{\sum_{j \in \Omega_\ell} \exp(\theta_j)} \leq \frac{\widetilde{\alpha}}{\kappa - \ell}$. We have,

$$
\begin{aligned}
&\sum_{j_{\ell-1} \in \Omega_\ell} \frac{\exp(\theta_{j_{\ell-1}})}{W - \sum_{k=j_1}^{j_{\ell-1}} \exp(\theta_k)} \\
=\ & \sum_{j_{\ell-1} \in \Omega_\ell} \frac{\exp(\theta_{j_{\ell-1}})}{W - \sum_{k=j_1}^{j_{\ell-2}} \exp(\theta_k) - \exp(\theta_{j_{\ell-1}})} \\
\geq\ & \frac{\sum_{j_{\ell-1} \in \Omega_\ell} \exp(\theta_{j_{\ell-1}})}{W - \sum_{k=j_1}^{j_{\ell-2}} \exp(\theta_k) - \left(\sum_{j_{\ell-1} \in \Omega_\ell} \exp(\theta_{j_{\ell-1}})\right)/|\Omega_\ell|} \\
=\ & \frac{\sum_{j_{\ell-1} \in \Omega_\ell} \exp(\theta_{j_{\ell-1}})}{\exp(\theta_i) + \exp(\theta_{i'}) + \sum_{j_{\ell-1} \in \Omega_\ell} \exp(\theta_{j_{\ell-1}}) - \left(\sum_{j_{\ell-1} \in \Omega_\ell} \exp(\theta_{j_{\ell-1}})\right)/|\Omega_\ell|} \\
=\ & \left(\frac{\exp(\theta_i) + \exp(\theta_{i'})}{\sum_{j_{\ell-1} \in \Omega_\ell} \exp(\theta_{j_{\ell-1}})} + 1 - \frac{1}{\kappa - \ell}\right)^{-1} \\
\geq\ & \left(\frac{\widetilde{\alpha}}{\kappa - \ell} + 1 - \frac{1}{\kappa - \ell}\right)^{-1} \\
=\ & \frac{\kappa - \ell}{\widetilde{\alpha} + \kappa - \ell - 1} = \sum_{j_{\ell-1} \in \Omega_\ell} \frac{\exp(\widetilde{\theta}_{j_{\ell-1}})}{\widetilde{W} - \sum_{k=j_1}^{j_{\ell-1}} \exp(\widetilde{\theta}_k)} ,
\end{aligned}
$$
(54)

(55)

(56)

where (54) follows from the Jensen's inequality and the fact that for any $c > 0$, $0 < x < c$, $\frac{x}{c-x}$ is convex in $x$. Equation (55) follows from the definition of $\widetilde{\alpha}_{i,i',\ell,\theta}$, (51), and the fact that $|\Omega_\ell| = \kappa - \ell$. Equation (56) uses the definition of $\{\widetilde{\theta}_j\}_{j \in S}$.

Consider $\{\Omega_{\widetilde{\ell}}\}_{2 \leq \widetilde{\ell} \leq \ell-1}$, $|\Omega_{\widetilde{\ell}}| = \kappa - \widetilde{\ell}$, corresponding to the subsequent summation terms in (53). Observe that $\frac{\exp(\theta_i) + \exp(\theta_{i'})}{\sum_{j \in \Omega_{\widetilde{\ell}}} \exp(\theta_j)} \leq \alpha/|\Omega_{\widetilde{\ell}}|$. Therefore, each summation term in equation (53) can be

lower bounded by the corresponding term where $\{\theta_j\}_{j\in S}$ is replaced by $\{\widetilde{\theta}_j\}_{j\in S}$. Hence, we have

$$\mathbb{P}_\theta\left[\sigma^{-1}(i), \sigma^{-1}(i') > \ell\right]$$

$$\geq \sum_{\substack{j_1\in S \\ j_1\neq i,i'}}\left(\frac{\exp(\widetilde{\theta}_{j_1})}{\widetilde{W} - \exp(\widetilde{\theta}_{j_1})} \cdots \sum_{\substack{j_{\ell-1}\in S \\ j_{\ell-1}\neq i,i', \\ j_1,\cdots,j_{\ell-2}}}\left(\frac{\exp(\widetilde{\theta}_{j_{\ell-1}})}{\widetilde{W} - \sum_{k=j_1}^{j_{\ell-1}}\exp(\widetilde{\theta}_k)} \sum_{\substack{j_\ell\in S \\ j_\ell\neq i,i', \\ j_1,\cdots,j_{\ell-1}}}\left(\frac{\exp(\theta_{j_\ell})}{W}\right)\cdots\right)\right)$$

$$\geq e^{-4b}\sum_{\substack{j_1\in S \\ j_1\neq i,i'}}\left(\frac{\exp(\widetilde{\theta}_{j_1})}{\widetilde{W} - \exp(\widetilde{\theta}_{j_1})} \cdots \sum_{\substack{j_{\ell-1}\in S \\ j_{\ell-1}\neq i,i', \\ j_1,\cdots,j_{\ell-2}}}\left(\frac{\exp(\widetilde{\theta}_{j_{\ell-1}})}{\widetilde{W} - \sum_{k=j_1}^{j_{\ell-1}}\exp(\widetilde{\theta}_k)} \sum_{\substack{j_\ell\in S \\ j_\ell\neq i,i', \\ j_1,\cdots,j_{\ell-1}}}\left(\frac{\exp(\widetilde{\theta}_{j_\ell})}{\widetilde{W}}\right)\cdots\right)\right)$$

$$= \left(e^{-4b}\right)\mathbb{P}_{\widetilde{\theta}}\left[\sigma^{-1}(i), \sigma^{-1}(i') > \ell\right]. \tag{57}$$

The second inequality uses $\frac{\exp(\theta_i)}{W} \geq e^{-2b}/\kappa$ and $\frac{\exp(\widetilde{\theta}_i)}{\widetilde{W}} \leq e^{2b}/\kappa$. Observe that $\exp(\widetilde{\theta}_j) = 1$ for all $j \neq i, i'$ and $\exp(\widetilde{\theta}_i) + \exp(\widetilde{\theta}_{i'}) = \widetilde{\alpha} \leq \lceil\widetilde{\alpha}\rceil = \alpha \geq 1$. Therefore, we have

$$\mathbb{P}_{\widetilde{\theta}}\left[\sigma^{-1}(i), \sigma^{-1}(i') > \ell\right] = \binom{\kappa-2}{\ell}\frac{\ell!}{(\kappa-2+\widetilde{\alpha})(\kappa-2+\widetilde{\alpha}-1)\cdots(\kappa-2+\widetilde{\alpha}-(\ell-1))}$$

$$\geq \frac{(\kappa-2)!}{(\kappa-\ell-2)!}\frac{1}{(\kappa+\alpha-2)(\kappa+\alpha-3)\cdots(\kappa+\alpha-(\ell+1))}$$

$$\geq \frac{(\kappa-\ell+\alpha-2)(\kappa-\ell+\alpha-3)\cdots(\kappa-\ell-1)}{(\kappa+\alpha-2)(\kappa+\alpha-3)\cdots(\kappa-1)}$$

$$\geq \frac{(\kappa-\ell)(\kappa-\ell-1)}{\kappa(\kappa-1)}\left(1-\frac{\ell}{\kappa+1}\right)^{\alpha-2}. \tag{58}$$

Claim (36) follows by combining Equations (57) and (58) and using the fact that $\alpha \leq 2e^{2b}$.
**Claim** (37): Define,

$$\widetilde{\alpha}_{\ell,\theta} \equiv \min_{i\in S}\min_{\ell'\in[\ell]}\min_{\substack{\Omega\in S\setminus\{i\} \\ :|\Omega|=\kappa-\ell'+1}}\left\{\frac{\exp(\theta_i)}{\left(\sum_{j\in\Omega}\exp(\theta_j)\right)/|\Omega|}\right\}. \tag{59}$$

Also, define $\alpha_{\ell,\theta} \equiv \lfloor\widetilde{\alpha}_{\ell,\theta}\rfloor$. Note that $\alpha_{\ell,\theta} \geq 0$ and $\widetilde{\alpha}_{\ell,\theta} \leq e^{2b}$. We denote the sum of the weights by $W \equiv \sum_{j\in S}\exp(\theta_j)$. Analogous to the proof of claim (36), we define the new set of parameters $\{\widetilde{\theta}_j\}_{j\in S}$:

$$\widetilde{\theta}_j = \begin{cases} \log(\widetilde{\alpha}_{\ell,\theta}) & \text{for } j = i, \\ 0 & \text{otherwise}. \end{cases} \tag{60}$$

Similarly define $\widetilde{W} \equiv \sum_{j\in S}\exp(\widetilde{\theta}_j) = \kappa - 1 + \widetilde{\alpha}_{\ell,\theta}$. Using the techniques similar to the ones used in proof of claim (36), we have,

$$\mathbb{P}_\theta\left[\sigma^{-1}(i) = \ell\right] \leq e^{4b}\mathbb{P}_{\widetilde{\theta}}\left[\sigma^{-1}(i) = \ell\right]. \tag{61}$$

Observe that $\exp(\widetilde{\theta}_j) = 1$ for all $j \neq i$ and $\exp(\widetilde{\theta}_i) = \widetilde{\alpha}_{\ell,\theta} \geq \lfloor\widetilde{\alpha}_{\ell,\theta}\rfloor = \alpha_{\ell,\theta} \geq 0$. Therefore, we have

$$\mathbb{P}_{\widetilde{\theta}}\left[\sigma^{-1}(i) = \ell\right] = \binom{\kappa-1}{\ell-1}\frac{\widetilde{\alpha}_{\ell,\theta}(\ell-1)!}{(\kappa-1+\widetilde{\alpha}_{\ell,\theta})(\kappa-2+\widetilde{\alpha}_{\ell,\theta})\cdots(\kappa-\ell+\widetilde{\alpha}_{\ell,\theta})}$$

$$\leq \frac{(\kappa-1)!}{(\kappa-\ell)!}\frac{e^{2b}}{(\kappa-1+\alpha_{\ell,\theta})(\kappa-2+\alpha_{\ell,\theta})\cdots(\kappa-\ell+\alpha_{\ell,\theta})}$$

$$\leq \frac{e^{2b}}{\kappa}\left(1-\frac{\ell}{\kappa+\alpha_{\ell,\theta}}\right)^{\alpha_{\ell,\theta}-1} \leq \frac{e^{2b}}{\kappa-\ell}. \tag{62}$$

Claim 37 follows by combining Equations (61) and (62).

**Claim (38):** Again, we construct a new set of parameters $\{\widetilde{\theta}_j\}_{j\in[d]}$ from the original $\theta$ using $\widetilde{\alpha}_{\ell,\theta}$ defined in (59):

$$\widetilde{\theta}_j = \begin{cases} \log(\widetilde{\alpha}_{\ell,\theta}) & \text{for } j \in \{i, i'\}, \\ 0 & \text{otherwise}. \end{cases} \tag{63}$$

Similarly define $\widetilde{W} \equiv \sum_{j\in S} \exp(\widetilde{\theta}_j) = \kappa - 2 + 2\widetilde{\alpha}_{\ell,\theta}$. Using the techniques similar to the ones used in proof of claim (36), we have,

$$\mathbb{P}_\theta\Big[\sigma^{-1}(i) = \ell_1, \sigma^{-1}(i') = \ell_2\Big] \leq e^{8b}\mathbb{P}_{\widetilde{\theta}}\Big[\sigma^{-1}(i) = \ell_1, \sigma^{-1}(i') = \ell_2\Big] \tag{64}$$

Observe that $\exp(\widetilde{\theta}_j) = 1$ for all $j \neq i, i'$ and $\exp(\widetilde{\theta}_i) = \exp(\widetilde{\theta}_{i'}) = \widetilde{\alpha}_{\ell,\theta} \geq \lfloor\widetilde{\alpha}\rfloor_{\ell,\theta} = \alpha_{\ell,\theta} \geq 0$. Therefore, we have

$$= \mathbb{P}_{\widetilde{\theta}}\Big[\sigma^{-1}(i) = \ell_1, \sigma^{-1}(i') = \ell_2\Big]$$

$$= \Bigg(\frac{\binom{\kappa-2}{\ell_2-2}\widetilde{\alpha}_{\ell,\theta}^2(\ell_2-2)!}{(\kappa-2+2\widetilde{\alpha}_{\ell,\theta})(\kappa-1+2\widetilde{\alpha}_{\ell,\theta})\cdots(\kappa-2+2\widetilde{\alpha}_{\ell,\theta}-(\ell_1-1))}$$

$$\frac{1}{(\kappa-2+\widetilde{\alpha}_{\ell,\theta}-(\ell_1-1))\cdots(\kappa-2+\widetilde{\alpha}_{\ell,\theta}-(\ell_2-2))}\Bigg)$$

$$\leq \frac{(\kappa-2)!}{(\kappa-\ell_2)!}\frac{e^{4b}}{(\kappa-2)(\kappa-1)\cdots(\kappa-\ell_1-1)(\kappa-\ell_1-1)\cdots(\kappa-\ell_2)}$$

$$\leq \frac{e^{4b}}{(\kappa-\ell_1-1)(\kappa-\ell_2)}. \tag{65}$$

Claim 38 follows by combining Equations (64) and (65).

## D  Proof of Theorem 4.2

Let $H(\theta) \in \mathcal{S}^d$ be Hessian matrix such that $H_{ii'}(\theta) = \frac{\partial^2 \mathcal{L}_{\text{RB}}(\theta)}{\partial\theta_i\partial\theta_{i'}}$. The Fisher information matrix is defined as $I(\theta) = -\mathbb{E}_\theta[H(\theta)]$. From lemma 2.1, $\mathcal{L}_{\text{RB}}(\theta)$ is concave. This implies that $I(\theta)$ is positive-semidefinite and from (22) its smallest eigenvalue is zero with all-ones being the corresponding eigenvector. Fix any unbiased estimator $\widehat{\theta}$ of $\theta \in \Omega_b$. Since, $\widehat{\theta} \in \mathcal{U}$, $\widehat{\theta} - \theta$ is orthogonal to $\mathbf{1}$. The Cramer-Rao lower bound then implies that $\mathbb{E}[\|\widehat{\theta} - \theta^*\|^2] \geq \sum_{i=2}^d \frac{1}{\lambda_i(I(\theta))}$. Taking supremum over both sides gives

$$\sup_\theta \mathbb{E}[\|\widehat{\theta} - \theta^*\|^2] \geq \sup_\theta \sum_{i=2}^d \frac{1}{\lambda_i(I(\theta))} \geq \sum_{i=2}^d \frac{1}{\lambda_i(I(\mathbf{0}))}. \tag{66}$$

In the following, we will show that

$$I(\mathbf{0}) = -\mathbb{E}_\theta[H(\mathbf{0})] \preceq \sum_{j=1}^n \sum_{a=1}^{\ell_j} \frac{m_{j,a} - \eta_{j,a}}{\kappa_j(\kappa_j-1)} \sum_{i<i'\in S_j} (e_i - e_{i'})(e_i - e_{i'})^\top \tag{67}$$

$$\preceq \max_{j,a}\{m_{j,a} - \eta_{j,a}\}L. \tag{68}$$

Using Jensen's inequality, we have $\sum_{i=2}^d \frac{1}{\lambda_i(I(\mathbf{0}))} \geq \frac{(d-1)^2}{\sum_{i=2}^d \lambda_i(I(\mathbf{0}))} = \frac{(d-1)^2}{\text{Tr}(I(\mathbf{0}))}$. From (67), we have $\text{Tr}(I(\mathbf{0})) \leq \sum_{j,a}(m_{j,a} - \eta_{j,a})$. From (68), we have $\sum_{i=2}^d 1/\lambda_i(I(\mathbf{0})) \geq (1/\max\{m_{j,a} - \eta_{j,a}\})\sum_{i=1}^d 1/\lambda_i(L)$. This proves the desired claim.

Now we are left to show claim (67). Consider a rank-breaking edge $e_{j,a}$. Using notations defined in lemma C.3, in particular Equation (43), and omitting subscript $\{j, a\}$ whenever it is clear from the context, we have, for any $i \in V(e_{j,a})$,

$$\frac{\partial^2 \mathbb{P}_\theta(e_{j,a})}{\partial^2\theta_i} = \begin{cases} \sum_\sigma\big(-A_\sigma B_\sigma e^{\theta_i} + A_\sigma(B_\sigma)^2 e^{2\theta_i} + A_\sigma C_\sigma e^{\theta_i}\big) & \text{if } i \in B(e_{j,a}) \\ \sum_\sigma\big(A_\sigma - 3A_\sigma B_{\sigma,i}e^{\theta_i} + A_\sigma C_{\sigma,i})e^{2\theta_i} + A_\sigma(B_{\sigma,i})^2 e^{2\theta_i}\big) & \text{if } i \in T(e_{j,a}), \end{cases} \tag{69}$$

and using (44), we have

$$\frac{\partial^2 \log \mathbb{P}_\theta(e_{j,a})}{\partial^2 \theta_i}\Big|_{\theta=0} = \begin{cases} ((C_\sigma - B_\sigma))_{\theta=0} & \text{if } i \in B(e_{j,a}) \\ \left(\frac{1}{m_{j,a}!}\sum_\sigma \left(C_{\sigma,i} - B_{\sigma,i} + (B_{\sigma,i})^2\right) - \left(\sum_\sigma \frac{B_{\sigma,i}}{m_{j,a}!}\right)^2\right)_{\theta=0} & \text{if } i \in T(e_{j,a}), \end{cases} \quad (70)$$

where $\sigma \in \Lambda_{T(e_{j,a})}$ and the subscript $\theta = 0$ indicates the the respective quantities are evaluated at $\theta = 0$. From the definitions given in (43), for $\theta = 0$, we have $B_\sigma - C_\sigma = \sum_{u=0}^{m-1}\frac{(r-u-1)}{(r-u)^2}$ and, $\sum_\sigma (B_{\sigma,i} - C_{\sigma,i})/(m!) = \frac{1}{m}\sum_{u=0}^{m-1}\frac{(m-u)(r-u-1)}{(r-u)^2}$. Also, $\sum_\sigma B_{\sigma,i}/(m!) = \frac{1}{m}\sum_{u=0}^{m-1}\frac{m-u}{r-u}$ and $\sum_\sigma (B_{\sigma,i})^2/(m!) = \frac{1}{m}\sum_{u=0}^{m-1}\left(\sum_{u'=0}^{u}\frac{1}{r-u'}\right)^2$. Combining all these and, using $\mathbb{P}_{\theta=0}[i \in T(e_{j,a})] = m/\kappa$ and $\mathbb{P}_{\theta=0}[i \in B(e_{j,a})] = (r-m)/\kappa$, and after following some algebra, we have for any $i \in S_j$,

$$-\mathbb{E}\left[\frac{\partial^2 \log \mathbb{P}_\theta(e_{j,a})}{\partial^2 \theta_i}\Big|_{\theta=0}\right]$$
$$= \frac{1}{\kappa}\left(m - \sum_{u=0}^{m-1}\frac{1}{r-u} - \frac{1}{m}\sum_{u=0}^{m-1}\frac{u(m-u)}{(r-u)^2} - \frac{1}{m}\sum_{u=0}^{m-2}\frac{2u}{r-u}\left(\sum_{u'>u}^{m-1}\frac{m-u'}{r-u'}\right)\right)$$
$$= \frac{m_{j,a} - \eta_{j,a}}{\kappa_j}, \quad (71)$$

where $\eta_{j,a}$ is defined in (11). Since row-sums of $H(\theta)$ are zeroes, (22), and for $\theta = 0$, all the items are exchangeable, we have for any $i \neq i' \in S_j$,

$$\mathbb{E}\left[\frac{\partial^2 \log \mathbb{P}_\theta(e_{j,a})}{\partial \theta_i \partial \theta_{i'}}\Big|_{\theta=0}\right] = \frac{m_{j,a} - \eta_{j,a}}{\kappa_j(\kappa_j - 1)}, \quad (72)$$

The claim (67) follows from the expression of $H(\theta)$, Equation (22).

To verify (71), observe that $(r-m)(B_\sigma - C_\sigma) + m(\sum_\sigma B_{\sigma,i}/(m!)) = m - \sum_{u=0}^{m-1}\frac{1}{r-u}$. And,

$$\frac{1}{m}\left(\sum_{u=0}^{m-1}\frac{m-u}{r-u}\right)^2 - \sum_{u=0}^{m-1}\left(\sum_{u'=0}^{u}\frac{1}{r-u'}\right)^2$$
$$= \sum_{u=0}^{m-1}\left(\frac{(m-u)^2}{m(r-u)^2} - \frac{m-u}{(r-u)^2}\right) + \sum_{0\le u<u'\le m-1}\left(\frac{2(m-u)(m-u')}{m(r-u)(r-u')} - \frac{2(m-u')}{(r-u)(r-u')}\right)$$
$$= \sum_{u=0}^{m-1}\frac{-u(m-u)}{m(r-u)^2} + \sum_{0\le u<u'\le m-1}\frac{-2u(m-u')}{m(r-u)(r-u')}.$$

## E   Tightening of Lemma C.3

Recall that $\mathbb{P}_\theta(e_{j,a})$ is same as probability of $\mathbb{P}_\theta[T(e_{j,a}) \succ B(e_{j,a})]$ that is the probability that an agent ranks $T(e_{j,a})$ items above $B(e_{j,a})$ items when provided with a set comprising $V(e_{j,a})$ items. As earlier, for brevity of notations, we omit subscript $\{j,a\}$ whenever it is clear from the context. For $m = 1$ or $2$, it is easy to check that all off-diagonal elements in hessian matrix of $\log \mathbb{P}_\theta(e)$ are non-negative. However, since number of terms in summation in $\mathbb{P}_\theta(e)$ grows as $m!$, for $m \ge 3$ the straight-forward approach becomes too complex. Below, we derive expressions for cross-derivatives in hessian, for general $m$, using alternate definition (sorting of independent exponential r.v.'s in increasing order) of PL model, where the number of terms grow only as $2^m$. However, we are unable to analytically prove that the cross-derivatives are non-negative for $m > 2$. Feeding these expressions in MATLAB and using symbolic computation, for $m = 3$, we can simplify these expressions and it turns out that they are sum of only positive numbers. For $m = 4$, with limited computational power it becomes intractable. We believe that it should hold for any value of $m < r$. Using (35), we need to check only for cross-derivatives for the case when $i \neq i' \in T(e_{j,a})$ or $i \in T(e_{j,a}), i' \in B(e_{j,a})$. Since, minimum of exponential random variables is exponential, we can assume that $|B(e_{j,a})| = 1$ that is $r = m + 1$. Define $\lambda_i \equiv e^{\theta_i}$. Without loss of generality, assume $T(e_{j,a}) = \{2, \cdots, m+1\}$

and $B(e_{j,a}) = \{1\}$. Define $C_x = \prod_{i=3}^{m+1}(1 - e^{-\lambda_i x})$. Then, using the alternate definition of the PL model, we have, $\mathbb{P}_\theta(e) = \int_0^\infty C_x(1 - e^{-\lambda_2 x})\lambda_1 e^{-\lambda_1 x}dx$. Following some algebra, $\frac{\partial^2 \log \mathbb{P}_\theta(e)}{\partial \theta_1 \partial \theta_2} \geq 0$ is equivalent to $A_1 \geq 0$, where $A_1 \equiv$

$$\left( \int C_x\big(xe^{-\lambda_1 x} - xe^{-\lambda x}\big)dx \right)\left( \int C_x xe^{-\lambda x}dx \right) - \left( \int C_x(e^{\lambda_1 x} - e^{-\lambda x})dx \right)\left( \int C_x x^2 e^{-\lambda x}dx \right),$$

where all integrals are from 0 to $\infty$ and, $\lambda \equiv \lambda_1 + \lambda_2$. Consider $A_1$ as a function of $\lambda_1$. Since $A_1(\lambda_1) = 0$ for $\lambda_1 = \lambda$, showing $\partial A_1/\partial \lambda_1 \leq 0$ for $0 \leq \lambda_1 \leq \lambda$ would suffice. Following some algebra, and using $\lambda_1 \leq \lambda$, $\partial A_1/\partial \lambda_1 \leq 0$ is equivalent to $A_2(\lambda_1) \equiv \left( \int_0^\infty C_x xe^{-\lambda_1 x} \right)/\left( \int_0^\infty C_x x^2 e^{-\lambda_1 x} \right)$ being monotonically non-decreasing in $\lambda_1$. To further simplify the condition, define $f^{(0)}(y) = 1/y^2$, $g^{(0)}(y) = 1/y^3$ and, $f^{(1)}(y) = f^{(0)}(y) - f^{(0)}(y + \lambda_3)$, and recursively $f^{(m-1)}(y) = f^{(m-2)}(y) - f^{(m-2)}(y + \lambda_{m+1})$. Similarly define $g^{(0)}, \cdots, g^{(m-1)}$. Using these recursively defined functions,

$$2A_2(\lambda_1) = \frac{f^{(m-1)}(\lambda_1)}{g^{(m-1)}(\lambda_1)},$$

$$\text{for } m = 3, \quad 2A_2(\lambda_1) = \frac{\lambda_1^{-2} - (\lambda_1 + \lambda_3)^{-2} - (\lambda_1 + \lambda_4)^{-2} + (\lambda_1 + \lambda_3 + \lambda_4)^{-2}}{\lambda_1^{-3} - (\lambda_1 + \lambda_3)^{-3} - (\lambda_1 + \lambda_4)^{-3} + (\lambda_1 + \lambda_3 + \lambda_4)^{-3}}.$$

Therefore, we need to show that $A_2(\lambda_1)$ is monotonically non-decreasing in $\lambda_1 \geq 0$ for any non-negative $\lambda_3, \cdots, \lambda_m$, and that would suffice to prove that the cross-derivatives arising from $i \in T(e_{j,a}), i' \in B(e_{j,a})$ are non-negative.

For cross-derivatives arising from $i \neq i' \in T(e_{j,a})$, define $B_x = \prod_{i=4}^{m+1}(1 - e^{\lambda_i x})e^{-\lambda_1 x}$. $\frac{\partial^2 \log \mathbb{P}_\theta(e)}{\partial \theta_2 \partial \theta_3} \geq 0$ is equivalent to $A_3 \geq 0$, where $A_3 \equiv$

$$\left( \int B_x(1 - e^{-\lambda_2 x})(1 - e^{-\lambda_3 x})dx \right)\left( \int B_x x^2 e^{-(\lambda_2 + \lambda_3)x}dx \right)$$

$$- \left( \int B_x(1 - e^{-\lambda_2 x})xe^{-\lambda_3 x}dx \right)\left( \int B_x(1 - e^{-\lambda_3 x})xe^{-\lambda_2 x}dx \right),$$

where all integrals are from 0 to $\infty$. For $m = 3$, using MATLAB we can show that both types of cross-derivatives are non-negative.