[Reviews · NeurIPS 2016]

Reviewer 1

Summary

In this article, the problem of aggregating ordinal data based on users preferences is studied. Plackett-Luce model is assumed for the ordering and a maximum likelihood estimator is presented. However, the computation of this estimator can be intractable in some case. The author propose the generalized ranking breaking principle to make the computation feasible. They provide a upper bound and a lower bound to support there approach. Some numerical experiment are also presented to illustrate.

Qualitative Assessment

I believe that this article is interesting and the original. However, the notations are very heavy so the article is very difficult to read. Section 3 is just unreadable, the goal is not explicit, there are references to supplementary materials and to theorem given after and the notation are given at the end without explanations.

Confidence in this Review

2-Confident (read it all; understood it all reasonably well)


Reviewer 2

Summary

The paper studies parameter estimate of Plackett-Luce (PL) model with partial ordered datasets from users. The traditional MLE suffers from combinatorial explosion of complexity in evaluating the log-likelihood function. On the other extreme, conversion of partial ordered data into pairwise comparison will suffer the dependence of samples or the lost of sample complexity. To achieve a tradeoff between such computational complexity and statistical precision, a novel generalized rank-breaking is proposed to bridge the gap between MLE and pairwise rank-breaking. Roughly speaking, it extracts maximal ordered partitions from data and only inputs a sub sample set whose top-set capacity is bounded by a prescribed M, for M=1 which meets the pairwise comparison sample. The proposed estimator is shown tractable and consistent, with a proved upper bound on the error rate in the finite sample regime. Numerical experiments show the tradeoff control by M on estimation error.

Qualitative Assessment

The consistency result seems to rely on the PL model to generate the data, how about other random utility models in the location family, say?

Confidence in this Review

2-Confident (read it all; understood it all reasonably well)


Reviewer 3

Summary

This paper considers the problem of rank aggregation. Given ordinal data about users’ preferences, the goal is to aggregate them in the form of partially ordered sets (a poset). Assuming the revealed preference is consistent, the poset can be represented as a directed acyclic graph. The authors propose a “generalized rank-breaking” algorithm, which is a unified framework which spans from the maximum likelihood estimator to pairwise rank-breaking. The MLE is consistent, but can be intractable and impractical. A common remedy is to use pairwise rank-breaking, which treats all pairwise comparisons as independent. However, this might lead to a statistically inconsistent estimator due to the ignored correlations. The proposed algorithm broadly does the following: Step 1) For each user j \in [n], extract a maximal ordered partition P_j of S_j which is consistent with her preference G_j; Step 2) Represent the extracted ordered partition by a directed hypergraph G_j(S_j, E_j) with e = (B(e), T(e)) \in E_j being a directed hyperedge; Step 3) Compute the probability of B(e) < T(e) in terms of parameters \theta_i (latent true utility); Step 4) Maximize the log-likelihood composed of the terms only with |T(e)| \leq M under the independent assumption. The authors show that finding an order-M rank-breaking estimate is a concave maximization with high probability, and that the estimate is consistent for sufficiently large M under the Plackett-Luce model. The main result of the paper suggests a high-probability upper bound on the achieved error of the estimate. This upper bound is optimal up to a logarithmic factor with respect to the Cramer-Rao bound when (a) the topology of the data is well-behaved (quantities defined on line 206 are finite), and (b) there is no limit on the computational power and M \to \infty.

Qualitative Assessment

Overall, the paper is well-organized: the process for generalized rank-breaking is clearly explained and the analysis provides an upper bound on the achieved error. It is also shown that this upper bound is nearly optimal under certain conditions, and the computational-statistical trade-off is also glimpsed. However, there are many parameters/quantities involved in the analysis so that sometimes it becomes not easy to follow and appreciate the authors’ arguments. I would like to suggest having an easy-to-find notation paragraph which readers can refer to, while reading. Also, there are several questions with regard to the contents of the paper: - Line 52: Can this generalized rank-breaking be applied beyond the Plackett-Luce model? - Line 98 – 99: I do not understand why none of the pairwise orderings can be used from G_j in the example. - Line 206: Does $\gamma_3$ also have an intuitive interpretation like $\gamma_1$ and $\gamma_2$? Also, is there any special reason you use $\nu$, not $\gamma_4$ to define the last quantity? Isn’t that related to the topology of the data? - Lines 244 – 251: I am a little bit confused. Lines 244 – 247 implies the upper bound is tight with respect to the lower bound when the topology of the data is well-behaved. However, Lines 248 – 251 seem to mean the opposite: Eq (13) gives a tighter lower bound when the topology is not well-behaved. Could you please clarify what I am misunderstanding for me? - Line 259: I am not following where in the analysis the increase of the error accompanied by the growth of m is predicted. Typoes: - Line 169: One -> On - Line 188: position of 'and'

Confidence in this Review

2-Confident (read it all; understood it all reasonably well)


Reviewer 4

Summary

This paper generalizes rank breaking from pairwise comparisons to more general partial orders under the PL model by allowing the user/analyst to tweak a knob that governs the trade off among consistency, accuracy and computational effort.

Qualitative Assessment

The main result is Thm 4.1 that gives a bound only O(log d) worse that number of parameters, which is interesting. The proof techniques extend from previous works to allow for the generalization which is novel contribution and could be useful in other analyses. The technical part of the paper it well written, as is about the related literature on rank breaking. However, I would suggest expanding some references in context of this work. e.g. lines 16, 30,88 talk about a bunch of references all grouped into respective umbrella buckets. Also, for readers not versed in definitions of consistency and 'topology of data' used in this literature, a definition in the beginning might have helped. The concavity portion seems over-emphasized IMO and could be shortened to make room for a smoother buildup. This is a theory paper and the real data experiments are merely proof of concept. It might have been a good idea to also present results on metrics used in practice (e.g. ndcg). A couple of minor language issues: line 19 (may be use 'fall back' instead of 'back-off' ? ); rephrase sentence in line 127 to cleanly define B(e) T(e); the para 248 to 251 does not read well to me, may be clarifying what topology means could help.

Confidence in this Review

2-Confident (read it all; understood it all reasonably well)


Reviewer 5

Summary

This paper considers the problem of aggregating partial rankings (provided by users as posets), under the Plackett-Luce data model. For general posets, computing the MLE is superexponential in the number of items. Prior work considered estimating using all pairwise comparisons in the data (treating these as independent), but this can be (statistically) inconsistent. Weighting (or only using some of) these comparisons is consistent [2, 3, 15], but is claimed to be inefficient. This work casts the comparisons in the data as a directed hypergraph, and drops comparisons with a user-specified max in-degree ("M"), which is seen as moving beyond pairwise comparisons. Theory is given showing that: this procedure is consistent, a user can trade "M" for a particular estimation error and computational cost, and a constrained minimax-type lower bound on the squared estimation error is available. Experiments confirm the theoretical predictions.

Qualitative Assessment

Novelty: - I understand the method, but I'm just a bit surprised that it does better (empirically) than using pairwise comparisons in an "intelligent" way, i.e., [3, 15]. Can you explain why? - Actually I'm a bit confused here. You write that [3, 15] are consistent (L94, 100), but write in your legends "inconsistent PRB" (Figs. 2, 3), and show that these methods behave inconsistently in these plots. Can you clarify? - Also, I wonder if your method really worth it? How long does "inconsistent PRB" take to run (we know your method's runtime from Fig. 2 left)? It does an OK job for a small number of samples in Fig. 2 middle, and maybe this is acceptable depending on the runtime. You should actually probably be plotting something like Pareto curves here. Regimes of complexity: Based on your choice of theta's (L254), you appear to be considering P-L distributions that are basically flat. Prior work (e.g., http://www.sciencedirect.com/science/article/pii/S0165489611000989) has pointed out that a ranking method's performance depends on the hardness of the underlying distribution. What is performance like under more peaked (i.e., easier) distributions? (Maybe these considerations also give you tighter bounds.) Related work: You may wish to be aware of work on the Mallows' model side of things (e.g., http://link.springer.com/article/10.1007%2Fs10458-013-9236-y), which has considered issues similar to yours in the past. Also, I wonder if parts of the active learning to rank literature are relevant here. Presentation: - I thought Sec. 1 was well-written. - I didn't see the point of Sec. 3, as it just sets up the main result. - Fig. 1: the shading of e1 and e2 here didn't make sense to me. - L184: where did "with high probability" come from? Remark 2.1 as stated in the main paper is deterministic. - Fig. 2: axis scales seem weird. - L51: there's no need for the "j" subscript, right? Aren't all users (partially) ranking the same ground set of items? - L79: what is "traditional" structure?

Confidence in this Review

2-Confident (read it all; understood it all reasonably well)